# Auditing the fairness of the US COVID-19 forecast hub's case prediction models

**Saad Mohammad Abrar**[1], **Naman Awasthi**[1], **Daniel Smolyak**[1],
**Nekabari Sigalo**[2], **Vanessa Frias Martinez**[1,2]

**1** Department of Computer Science, University of Maryland, College Park, Maryland, United States of America, **2** College of Information and UMIACS, University of Maryland, College Park, Maryland, United States of America, **3** Abt Global, Rockville, Maryland, United States of America

**Data availability statement:** The datasets supporting the conclusions of this article are available in the following repositories: 1. All COVID-19 Forecasts are available at:

## Abstract

The US COVID-19 Forecast Hub, a repository of COVID-19 forecasts from over 50 independent research groups, is used by the Centers for Disease Control and Prevention (CDC) for their official COVID-19 communications. As such, the Forecast Hub is a critical centralized resource to promote transparent decision making. While the Forecast Hub has provided valuable predictions focused on accuracy, there is an opportunity to evaluate model performance across social determinants such as race and urbanization level that have been known to play a role in the COVID-19 pandemic. In this paper, we carry out a comprehensive fairness analysis of the Forecast Hub model predictions and we show statistically significant diverse predictive performance across social determinants, with minority racial and ethnic groups as well as less urbanized areas often associated with higher prediction errors. We hope this work will encourage COVID-19 modelers and the CDC to report fairness metrics together with accuracy, and to reflect on the potential harms of the models on specific social groups and contexts.

## Introduction

### The US COVID-19 forecast hub: Development and context

The US COVID-19 Forecast Hub was founded in 2020 and serves as a *"central repository of COVID-19 forecasts from over 50 independent research groups"* [1]. Participant research groups submit county, state and national US COVID-19 forecasts with a standardized format; and the Forecast Hub provides an interactive visualization tool to help decision makers and the general public analyze weekly predictions for COVID-19 hospitalizations, cases and deaths. The standardized predictions collected from all research groups, as well as the predictions for an ensemble model that brings all individual predictions together, are also shared with the Centers for Disease Control and Prevention (CDC) who uses these results for their official COVID-19 communications [2].

Over the past four years, numerous research groups from both academia and industry have focused on developing models to forecast COVID-19 cases, hospitalizations and deaths in the United States. The COVID-19 Forecast Hub [3] has been instrumental in collating these efforts. These models vary in approach, ranging from deep learning methods [4–7] to

https://github.com/reichlab/covid19-forecast-hub/tree/master 2. All COVID-19 Ground Truth Data are available at: https://github.com/CSSEGISandData/COVID-19 3. All Race and Ethnicity, Population Data are available at: https://www.census.gov/data/tables/time-series/demo/popest/2020s-counties-detail.html 4. The Urban-Rural Classification Scheme for Counties: https://www.cdc.gov/nchs/data_access/urban_rural.htm#2013_Urban-Rural_Classification_Scheme_for_Counties 5. All Health Outcome Datasets are provided at: https://data.cdc.gov/500-Cities-Places/PLACES-Local-Data-for-Better-Health-Place-Data-202/eav7-hnsx/about_data.

**Funding:** This work has been funded with National Science Foundation grants, NSF #1750102 and NSF #2210572. The funders had no role in study design, data collection and analysis, decision to publish, or preparation of the manuscript.

**Competing interests:** The authors have declared that no competing interests exist.

compartmental models [8,9], statistical models [10,11], or combinations of these via ensemble models [3,12]. COVID-19 forecast models are usually trained with historical data (e.g., past cases or hospitalizations) together with other contextual information such as human mobility data. Human mobility data has been used in the past to model and characterize human behaviors in the built environment [13–17], to support decision making for socio-economic development [18–22], for public safety [23,24], as well as during epidemics and disasters [25–30]. During the COVID-19 pandemic, human mobility has also played a central role in driving decision making, and more than 50% of the Forecast Hub models have incorporated mobility data into their prediction models, acknowledging the impact of human movement on virus propagation [5–7,31–33].

## Fairness in COVID-19 prediction models: A critical gap

The US COVID-19 Forecast Hub has been, and continues to be, a critical centralized resource to promote transparent decision making. While the Forecast Hub has made significant contributions through its accuracy-focused predictions at different spatial granularities (*e.g.,* county or state), there is an opportunity to expand its evaluation framework to examine how prediction performance varies across social determinants like race, ethnicity and urbanization levels that have been shown to play an important role in COVID-19, including race, ethnicity and rurality [34,35].

The pandemic has highlighted existing disparities in healthcare, with significant differences in COVID-19 infection rates, hospital admissions, and deaths among different racial and ethnic groups as well as across the urban-rural spectrum [35,36]. These disparities risk being perpetuated in model predictions if not adequately addressed. Diverse prediction performance across social determinants - for example, higher prediction errors for a given minority race or ethnicity - could negatively impact resource allocation and intervention decisions e.g., hospital beds or stay-at-home orders, given that the CDC appears to be using the Forecast Hub predictions for official communications that subsequently inform policy decisions [2]. Given the urgent need for rapid pandemic response modeling, initial Forecast Hub efforts necessarily focused on developing accurate predictions. As these models continue to inform CDC communications and policy decisions, incorporating fairness analyses could further enhance their utility for equitable resource allocation and intervention planning across diverse communities [37].

There are many reasons why the COVID-19 prediction performance can be different across social determinants such as race, ethnicity or urbanization levels. The Forecast Hub's COVID-19 prediction models are trained on datasets containing COVID-19 statistics for hospitalizations, cases or deaths. Given the unprecedented scale and urgency of the pandemic, data collection faced several challenges [34,38]. For example, a lack of consistency in reporting race and ethnicity across jurisdictions, has generated a lot of missing racial data. That data is often excluded due to its incompleteness, potentially affecting the actual total hospitalizations, cases or deaths for minority race and ethnicity groups which might be less reported. In addition, there are occasions where the race is reported by the medical staff instead of being self-reported, which is the most accurate source and prevents errors [39]. For example, the CDC reports that the latest research on race and Hispanic origin misclassification on COVID-19 death certificates shows that deaths are underreported by 33% for non-Hispanic American Indian or Alaska Natives, by 3% for non-Hispanic Asian or Pacific Islanders, and by 3% for Hispanic decedents [40]. Testing availability and access varied across communities, with some minority groups experiencing more limited access, such as Latino communities [41], thus

affecting the accuracy of the overall COVID-19 statistics, with under-reporting bias perpetuating the invisibility of racial and ethnic minorities in general COVID-19 statistics. A similar effect has been observed in rural counties and states, with rural areas associated to lower testing rates, thus disproportionately detecting fewer cases of COVID-19 in these regions [35].

To exacerbate this situation even more, COVID-19 prediction performance across social determinants can also be affected by additional datasets used in the training of some of the COVID-19 prediction models. Specifically, around 50% of the Forecast Hub's models use human mobility data from Safegraph [42], Apple [43] or Google [44] among others, to complement COVID-19 predictions (see Fig 2A). Human mobility data can characterize origin-destination trips, visits to specific points of interest (POI), or the volumes of different types of trips (*e.g.,* car vs. public transit). Research has shown that mobility data can improve the prediction accuracy of COVID-19 cases, deaths and hospitalizations [33,45,46]. Nevertheless, researchers have also identified that mobility data suffers from sampling bias across race and age groups [31] with, for example, elder and Black communities being less represented [47]. Similarly to the COVID-19 case under-reporting bias, mobility data sampling bias could also affect the fairness of COVID-19 predictions across social groups.

In this paper, we propose - to the best of our knowledge - the first thorough fairness analysis of the COVID-19 prediction models in the Forecast Hub. Specifically, we focus on COVID-19 case prediction models at the county level, since these are closer to local realities and allow for more actionable decision making than state-level predictions. We use error parity as a measure of group fairness [48] *i.e.,* lack of fairness in our context is associated with significantly different error distributions across two social determinants: race or ethnicity and urbanization level. Accurately revealing differences across racial and ethnic groups would require access to county-level COVID-19 case data stratified by race or ethnicity, which would allow us to compare predicted versus actual case county statistics for each racial and ethnic group. Due to the complexity and scope of pandemic data collection efforts, many counties in the US faced significant challenges in collecting comprehensive demographic data [49]. Hence, to be able to carry out a fairness analysis of the Forecast Hub's COVID-19 prediction models, we propose a regression analysis to evaluate the associations between prediction errors in a given county and the race and ethnicity distributions for that county, while controlling for underlying health conditions and age groups. A similar regression analysis is proposed for the urbanization levels.

Additionally, to support researchers in the Forecast Hub, we also investigate how group fairness metrics for race, ethnicity and urbanicity levels change across model characteristics such as model type (*e.g.,* deep learning versus statistical), training data (*e.g, with or without mobility data*), lookaheads (*e.g.,* predicting cases for next week versus in four weeks) or pandemic phases. Finally, we also describe a dashboard that we have designed to allow decision makers and researchers explore *fairness nutritional cards* for each Forecast Hub model [50]. To sum up, the main contributions of this paper are:

- We present a thorough fairness analysis of the CDC Forecast Hub's COVID-19 county case prediction models across race, ethnicity and urbanization levels. Our research shows statistically significant differences in predictive errors with some minority racial and ethnic groups as well as less urbanized areas associated with significantly higher errors than the majority White race, while controlling for underlying health conditions, age groups and state.
- We carry out interaction analyses identifying differences in performance across racial/ethnic groups and urbanicity levels with respect to COVID-19 prediction type models, COVID-19 datasets, prediction lookaheads and COVID-19 phases, while controlling

for underlying health conditions and age groups. Our results show significant prediction performance differences for certain minority groups and less urbanized areas, when compartmental or statistical models are used. On the other hand, short-term forecasting and certain pandemic phases with higher case volumes are also associated with higher prediction performance differences for certain minority racial and ethnic groups as well as for less urbanized areas.

• We present a dashboard where researchers and decision makers at the CDC and beyond will be able to explore fairness nutritional cards per individual model across race, ethnicity and urbanization level, and how fairness might vary across model and data characteristics.

## Materials and methods

### Data sources and variables

**COVID-19 forecast data.** For the purpose of our study, we focus exclusively on the weekly, county-level COVID-19 case predictions publicly available from the COVID-19 Forecast Hub across all US counties [1]. We focus on county-level forecasts because these are closer to local realities and allow for more actionable decision-making than state-level predictions. On the other hand, since hospitalizations and deaths are only available at the state level [3], we focus on COVID-19 case predictions. The weekly incidence predictions in the Forecast Hub are uploaded by participating teams and defined as the newly anticipated COVID-19 cases per county within the following *epidemiological week*, extending from Sunday to Saturday. We use the weekly forecasts during the period from July 2020 to October 2022.

The hub's data repository offers both point forecasts and quantile-based probabilistic forecasts. Our study employs the latter, leveraging the seven provided quantiles ([0.025, 0.100, 0.250, 0.500, 0.750, 0.900, 0.975]) to gain insights into the uncertainty ranges and confidence intervals posited by the forecasting models. From the entire cohort of models and teams contributing to the Forecast Hub, we selected 36 teams that met our inclusion criteria: they provided comprehensive quantile forecasts throughout our period of analysis and they submitted predictions at the county-level. A Gantt chart depicting the specific quantile forecasts used to evaluate each model is shown in Fig 1 in S1 Appendix.

**Sensitive attributes.** To empirically evaluate fairness in COVID-19 forecasts, we measure error parity across two sensitive demographic attributes: (1) RACE AND ETHNICITY population composition, and (2) county-level URBANIZATION LEVEL classification. For this analysis, we use two primary data sources: the American Community Survey (ACS) [51] for demographic information and the CDC urbanization classification for county-level urban-rural designations [52]. In our analysis, we focus on Asian, Black, Hispanic, and White racial/ethnic groups while using White as the reference category (see Fig 1A for the racial and ethnic distribution across the 3,067 counties considered for this study). Other racial categories, including American Indian/Alaska Native (AIAN), Native Hawaiian/Pacific Islander (NHPI), and multi-racial groups, were excluded from our primary analysis due to limited variable distribution. As Table 1 shows, 97.69% of counties have NHPI populations under 1%, and 79.03% have AIAN populations under 1%, providing insufficient variation for meaningful regression analysis. In contrast, between 1.21%–66.22% of the counties have less than 1% of Asian, Black, or Hispanic population, providing adequate statistical power for detecting potential effects in our regression analysis. Furthermore, Asian populations represent a key demographic group in metropolitan and suburban areas where COVID-19 impacts were particularly pronounced, making their inclusion crucial for understanding prediction accuracy in these important

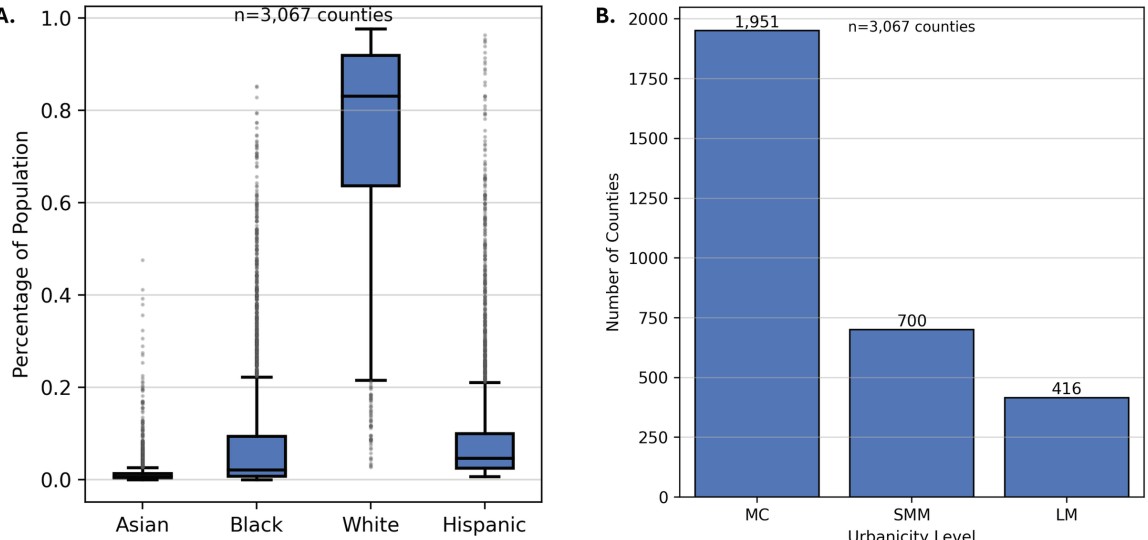

**Fig 1. Distribution of the sensitive attributes (race/ethnicity and urbanization level) across the 3,067 counties in US considered for this study.** (A) Distribution of racial demographics across US counties, (B) distribution of counties by urbanicity level.

**Table 1. County-level demographic distribution statistics.**

| Racial/Ethnic Group | Median (%) | Mean (%) | Counties > 1% | % Counties > 1% |
|---|---|---|---|---|
| White | 83.09 | 75.64 | 3,067/3,067 | 100.00 |
| Hispanic | 4.64 | 9.88 | 3,030/3,067 | 98.79 |
| Black | 2.14 | 8.88 | 2,019/3,067 | 65.83 |
| Asian | 0.70 | 1.53 | 1,036/3,067 | 33.78 |
| American Indian/Alaska Native | 0.40 | 2.00 | 643/3,067 | 20.97 |
| Native Hawaiian/Pacific Islander | 0.12 | 0.24 | 71/3,067 | 2.31 |

**Notes:** Statistics are computed across all 3,067 U.S. counties included in our analysis. "Counties > 1%" shows the number of counties where the group comprises more than 1% of the population.

contexts. Further details about this selection, and exclusion of some counties can be found in the S1 Appendix in Sect 1.1.

On the other hand, the CDC urban-rural classification scheme classifies counties into six urbanization levels [52], from highly urban (1) to rural (6). For this paper, we group them into three labels: Large Metropolitan areas (**LM**, which correspond to codes 1 and 2), Small and Medium Metropolitan (**SMM**, codes 3 and 4) and Micropolitan and Non-core areas (**MC**, codes 5 and 6). This grouping ensures sufficient sample sizes and variations within each category for robust statistical analysis, given the uneven distribution of US counties across urbanization levels (see Fig 1B). At the same time the three-level classification provides a clearer narrative about urban-rural disparities while maintaining meaningful distinctions in population density and healthcare infrastructure.

**Model-data characteristics.** Given our interest in understanding how group fairness metrics for race, ethnicity and urbanization might change across model and data characteristics, we break down the prediction performance and fairness analyses across four aspects:

- **Model Type**: Based on information reported in the papers associated to each of the 36 predictive models, we have manually classified them into five categories, namely: Statistical, Compartmental, Deep Learning, Baseline, and Ensemble (see Fig 2A for model statistics).

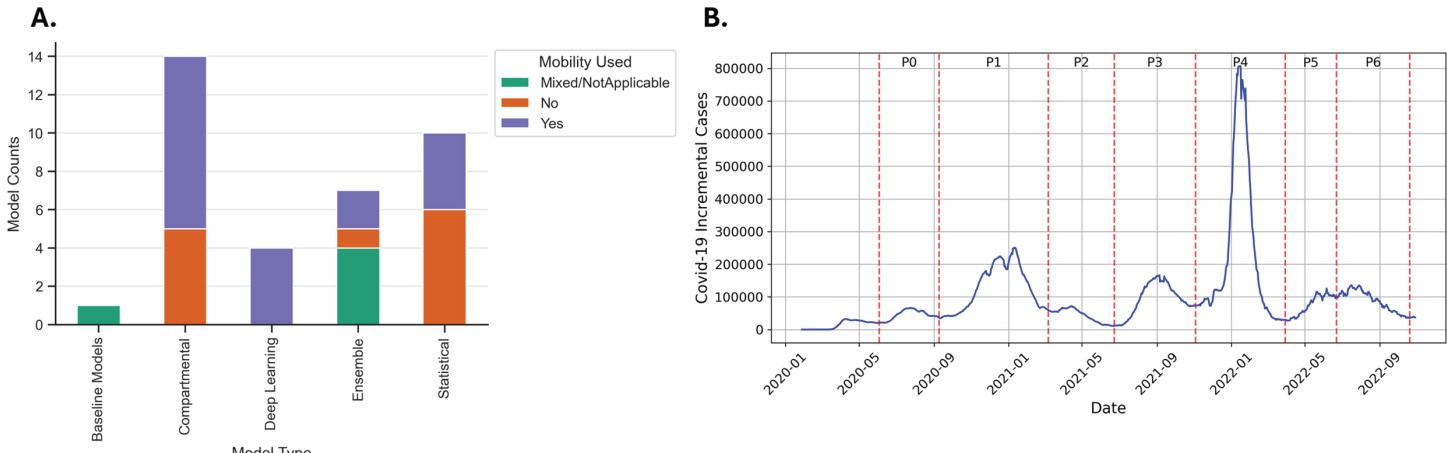

**Fig 2. Prediction fairness will be evaluated across types of models, training datasets (mobility), lookaheads and phases.** (A) Model types and mobility used counts (B) phase demarcation where P0 represents Phase 0.

This classification aims to discern the potential influence of model typologies on forecast performance and to identify any systematic biases inherent to specific modeling approaches (Sect 1.2 and Table 1 in S1 Appendix provide more details about the taxonomy).

- **Mobility Used**: Based on information reported in the papers for each predictive model analyzed, we distinguish between models that integrate mobility data and those that do not. The main objective of this feature is to allow us to explore the effect of mobility data on forecast accuracy and the possible introduction of biases resulting from this additional data dimension. Fig 2A displays a categorization of the forecasting models, differentiated by both the type of model and the incorporation of mobility data, as quantified by their respective counts. *Mixed models* are COVIDHub Ensemble models (like COVIDhub CDC-ensemble) that combine models that use (or not) mobility data in their training.

- **Lookahead**: We use forecasts ranging from 1 week to 4 weeks (a.k.a lookaheads), allowing us to evaluate differences in the predictive accuracy of the hub's models across race, ethnicity and the rural-urban spectrum for short and medium-term horizons: at 7, 14, 21 and 28 days.

- **Phase**: COVID-19 case numbers differ a lot across pandemic stages. To understand whether phases play a role in model fairness across race, ethnicity and urbanization level, we divide the period under study into seven phases, identified based on the presence of valleys and peaks in the volume of COVID-19 cases (see Fig 2B). A more detailed explanation of the phase identification process can be found in Sect 1.3 in the S1 Appendix.

**Control variables.** Our analysis controls for potential confounding factors by incorporating county-level demographic and health variables in our analyses. Specifically, we include the proportion of residents aged 65 and above (sourced from ACS [51]) to account for age-related COVID-19 risk variations. To control for underlying health conditions known to affect COVID-19 susceptibility and severity we also utilize age-adjusted prevalence of nine key comorbidities [53]: high blood pressure, cancer, diabetes, obesity, stroke, chronic obstructive pulmonary disease (COPD), Chronic kidney disease (CKD), current asthma, and coronary heart disease (CHD) extracted from the CDC PLACES [54] dataset. These specific health conditions were selected based on extensive epidemiological evidence linking them to increased COVID-19 severity and mortality [55].

## Analytical approach

To systematically evaluate fairness across race, ethnicity, and urbanization levels in COVID-19 case predictions from the Forecast Hub, we developed a three-step analytical approach that combines error calculation, fairness assessment across sensitive attributes, and interaction analysis with model-data characteristics, while controlling for relevant demographic and health factors.

**Step One: Calculating Pinball Loss.** We focus on county error parity as a measure of fairness. Hence, we first need to compute the weekly forecast error at the county level for all the counties in the US. To evaluate the accuracy of COVID-19 case forecasts, and given that forecasts in the Forecast Hub are uploaded by teams as quantiles, we employ the pinball loss metric (PBL). This metric quantifies the error of a probabilistic forecast by measuring the distance between observed values and the predicted quantiles, penalizing over- or under-estimation asymmetrically to reflect the actual cost of errors in the prediction. The pinball loss $L_\tau(y,f)$ is represented as: $(\tau - 1) \cdot (y - f)$, if $y < f$ and as $\tau \cdot (f - y)$, if $y \geq f$, where $L_\tau(y,f)$ denotes a county's pinball loss for a given quantile $\tau$, $y$ is the observed value i.e., reported number of cases or ground truth extracted from the JHU CSSE COVID-19 case counts dataset [56], and $f$ is the forecasted value at quantile $\tau$. For our analysis, we use the average county PBL, computed across the set of 7 quantiles: $\overline{PBL_\tau} = \frac{1}{7} \sum_{i=1}^{7} L_\tau(y_i, f_i)$, and normalized by the county population.

**Step Two: Fairness across Sensitive Attributes.** Once weekly average PBLs have been computed per county, we aim to evaluate associations between county prediction errors and the two protected attributes, RACE AND ETHNICITY and URBANIZATION LEVEL, using regressions. Coefficient analyses will enable us to identify cases in which error parity is violated, pointing to a lack of prediction fairness i.e., significantly different errors across racial or ethnic groups or across urbanization levels.

Associating counties, and their PBL errors, to URBANIZATION LEVEL is straight forward using the CDC urban-rural classification scheme [52] that associates a county with a given urbanization level. On the other hand, associating counties, and their PBL prediction errors, with race and ethnicity would require access to race-stratified predictions. However, due to systemic data collection challenges during the pandemic, the race-stratified COVID-19 case data necessary to compute race-stratified prediction errors were not collected, hence limiting the predictions provided by the Forecast Hub to county level predictions across all racial and ethnic groups [49]. Next, we describe how we proceed in the evaluation of the relationship between forecast errors and sensitive attributes race, ethnicity and urbanization level.

**Race and Ethnicity Analysis.** To evaluate whether there exist differences between PBL error distributions across racial and ethnic groups, we conduct a regression analysis where county PBL errors are regressed against the racial and ethnic percentage distributions within each county (with White race as the reference group). In the regression, we control for underlying health conditions, percentage of the population aged 65 and above, the state and data-model characteristics (i.e., type of predictive model, mobility data, lookahead and phase). An analysis of the resulting regression coefficients can provide insights into how different racial compositions might be positively or negatively associated with forecast accuracy, potentially identifying unfairly higher errors for certain racial or ethnic groups. The main effects regression model, **Model-1**, can be formally defined as:

$$PBL_{c,t} = \alpha + \sum_i \beta_i \cdot Race_{i,c} + \sum_{j,k} \gamma_{j,k} \cdot DM_{j,k,c,t} + \sum_l \sigma_l \cdot HO_{l,c} + \theta \cdot age65_c + \sum_s \lambda_s \cdot State_{s,c} + \epsilon_{c,t} \quad (1)$$

where:

- $PBL_{c,t}$: The dependent variable representing prediction error for county c at time t.
- $\alpha, \epsilon_{c,t}$: The intercept term and the error terms for county c at time t, respectively.
- $\beta_{\text{Asian}}, \beta_{\text{Black}}, \beta_{\text{Hispanic}}$: Relationship between each race category (Asian, Black, and Hispanic) and $PBL$, with White as the reference category.
- $\gamma_{jk}$: Relationship between each model-data characteristic j with corresponding category k and the PBL, with $j \in$ {Lookahead, Phase, Model Type, Mobility Used}. For example, for the characteristic Lookahead, $k \in (14, 21, 28)$ (7-days reference group). Reference groups for each model-data characteristic are: 7-days, phase 0, compartmental model and no mobility data used.
- $\sigma_l$: Relationship between age-adjusted health outcomes ($HO$) and the PBL, with $l \in$ {BPHIGH, CANCER, DIABETES, OBESITY, STROKE, COPD, KIDNEY, CASTHMA, CHD}. For example, $\sigma_{Asthma}$ represents the effect of asthma prevalence in a given county on the PBL.
- $\theta$: Relationship between the percentage of population aged 65 and older in a given county and the $PBL$.
- $\lambda_s$: Individual state s fixed effect on the error distribution, with $s \in$ {List of States in US}.

**Urbanization Analysis.** For the Urbanization Level we replace the Race percentage variables in Eq 1 with the three urbanization levels and we refer to this as **Model-2**.

$$PBL_{c,t} = \alpha + \sum_i \beta_i \cdot Urb_{i,c} + \sum_{jk} \gamma_{jk} \cdot DM_{j,k,c,t} + \sum_l \sigma_l \cdot HO_{l,c} + \theta \cdot age65_c + \sum_s \lambda_s \cdot State_{s,c} + \epsilon_{c,t} \quad (2)$$

where, $Urb_{i,c}$ where, $Urb_{i,c}$ represents the urbanization leveli for county c, with $i \in SMM, MC$, with LM being the reference category.

**Step Three: Interaction with Model-Data Characteristics.** Building upon the main effects models, we are also interested in looking into whether the fairness metrics across sensitive attributes change when model and/or data characteristics are considered. In other words, we evaluate if the relationship between county forecast errors and their racial/ethnic and urbanization groups changes when model and data characteristics are taken into account. Next, we describe the methodological approach for each sensitive attribute.

**Race and Ethnicity Analysis.** To evaluate whether the relationship between PBL error distributions and race/ethnicity changes across model or data characteristics, we regress the county PBL errors against racial and ethnic percentages for that county (with the White group as a baseline) while adding interaction terms between race/ethnicity and each model-data characteristic, and while controlling for age-adjusted Health Outcome prevalence, the percentage of individuals in the county who are 65+, and the county state. Since we are interested in four model-data characteristics, namely model type, use of mobility data, lookahead and phase, we create for separate regressions (**Model-1a, -1b, -1c and -1d**, respectively) to account for both main and interaction effects for each model-data characteristic:

$$PBL_{c,t} = \alpha + \sum_i \beta_i \cdot Race_{i,c} + \sum_{jk} \gamma_{j,k} \cdot DM_{j,k,c,t} + \sum_{i,k} \delta_{ijk} \cdot (Race_{i,c} \times DM_{j,k,c,t})$$
$$+ \sum_l \sigma_l \cdot HO_{l,c} + \theta \cdot age\_65_c + \sum_s \lambda_s \cdot State_{s,c} + \epsilon_{c,t} \quad (3)$$

where $\delta_{ijk}$ represents the relationship between the PBL and changes in a given race percentagei and model– data characteristic category j, k with respect to their reference groups (e.g., $\delta_{Hispanic,Lookahead,14-days}$ helps us evaluate how the PBL changes for a 1% increase in $i = Hispanic$

county population with respect to the White reference group, with $j$ = *Lookahead* and $k$ = 14 – *days* with respect to reference category 7-days). All other terms remain the same as Eq 1. To be able to evaluate significant changes in the error distribution (PBL) between a minority race and the White reference group for a given model-data characteristic category $j, k$, we will examine the relative effect computed as $\beta_i + \delta_{ijk}$.

**Urbanization Analysis.** Similarly to the race and ethnicity analysis, we construct four regression models (**Model-2a,-2b, -2c and -2d**) that mirror the race/ethnicity interaction models, but replace racial percentages with urbanization categories. Using Large Metropolitan (LM) areas as the reference group, we examine interactions between urbanization levels (SMM and MC) and each model-data characteristic. The regression equation takes the form:

$$PBL_{c,t} = \alpha + \sum_i \beta_i \cdot Urb_{i,c} + \sum_{jk} \gamma_{jk} \cdot DM_{j,k,c,t} + \sum_l \sigma_l \cdot HO_{l,c}$$
$$+ \sum_{i,k} \delta_{ijk} \cdot (Urb_{i,c} \times DM_{j,k,c,t}) + \theta \cdot age\_65_c + \sum_s \lambda_s \cdot State_{s,c} + \epsilon_{c,t} \tag{4}$$

where $Urb_{i,c}$ represents the urbanization level $i$ for county $c$, and the interaction term $\delta_{ijk}$ captures the relationship between a given urbanization level and model– data characteristic category $j, k$ with respect to their reference groups, on the PBL. We will examine the relative effect of a given urbanization level $i$ on the PBL (with respect to LM) and for a given model– data characteristic category $j, k$ computing the relative effect as $\beta_i + \delta_{ijk}$.

Table 2 provides a comprehensive overview of our regression framework, detailing the dependent and independent variables for each model. The table systematically presents our main effects models (Model-1 and Model-2) and their corresponding interaction models, along with the control variables and model specifications used throughout our analysis.

## Model selection and evaluation

Initial exploratory analysis of the Pinball Loss norm (PBL) distribution revealed several characteristics that influenced our model selection. The original distribution exhibited extreme right-skewness with a heavy tail, through quantile analysis (see Sect 2.1 and Fig 2 in S1 Appendix). These extreme values, while valid measurements, can exert excessive leverage in regression models and potentially obscure patterns in the majority of the data. Therefore, we made the methodological choice to trim the top 1% values, preserving 99% of our observations while substantially improving the stability of the model. We also consider squared root transformation during model evaluation. Given these distributional characteristics - which violate basic linear regression assumptions- and the strictly positive nature of our dependent variable, we tested both Gamma and Gaussian GLM families with log-link and identity-link functions.

Initial analysis revealed substantial multicollinearity among independent variables in the GLM models, particularly between health variables and racial demographics. To address this, we employed an iterative variable selection process using the Generalized Variance Inflation Factor (GVIF) [57,58]. Variables were retained only if their adjusted GVIF values (calculated as $(GVIF^{\frac{1}{2 \times Df}})$, where Df represents degrees of freedom) fell below a threshold of 2. Importantly, we proceeded with our analysis only after confirming that our variables of interest - race, ethnicity, urbanization level and data-model characteristics - all demonstrated adjusted GVIF values below this threshold, ensuring the reliability of our primary coefficient estimates.

Using diagnostic plots, residual patterns, and model performance metrics (pseudo-$R^2$ values) for model evaluation, we identified the Gaussian GLM with log-link function applied

to square root transformed data (with 1% trimming) as the best fit. Multicollinearity analyses and model diagnostics for these GLMs are discussed in depth in Sects 2.2 and 2.3 in S1 Appendix.

In the next section, we evaluate the relationship of race, ethnicity and urbanization levels with the forecast error (PBL) distribution via Model-1 and Model-2, which we rename to GLM-1 and GLM-2. We also evaluate how that relationship is modulated when interaction terms between race, ethnicity, urbanization levels and the four model and data characteristics are incorporated into the model i.e., Model-1a through Model-1d and Model-2a through Model-2d that we rename as GLM-1a through GLM-1d and GLM-2a through GLM-2d (see Table 2 for model summary). Following the Results section, we will present hypotheses for the reasons behind these findings in the Discussion section.

## Results

### Fairness of COVID-19 case predictions across race and ethnicity (GLM-1)

Based on Eq 1, the generalized linear model (GLM-1) characteristics are illustrated in Table 3. The table provides insights into the relationship between race, ethnicity and the PBL while controlling for the impact of various data-model features, health outcomes and state fixed effects on prediction accuracy. As a note, some of the health outcome covariates were left out due to inflated GVIF.

For the race and ethnicity variables, the regression results indicate that for every 1% increase in a county's Hispanic population (with respect to the reference group White) the prediction errors increased by approximately $(1.216 - 1) * 100\% = 21.6\% (\beta = 0.196, \exp(\beta) = 1.216, p < 0.001)$. In contrast, regions with larger Asian populations showed markedly

**Table 2. Summary of generalized linear models and their specifications.**

| Model | Sensitive Attributes | Interaction Terms | Control Variables |
|---|---|---|---|
| *Main Effects Models* | | | |
| GLM-1 | Race/Ethnicity | None | Health Outcomes |
| | (% Black, Hispanic, Asian) | | Data-Model Characteristics |
| | % White (ref.) | | State Fixed Effects |
| GLM-2 | Urbanicity | None | Health Outcomes |
| | (SMM, MC), | | Data-Model Characteristics |
| | LM (ref.) | | State Fixed Effects |
| *Main + Interaction Effects: Race Interaction Models* | | | |
| GLM-1a | Race/Ethnicity | Race × Lookahead | Same as GLM-1 |
| GLM-1b | Race/Ethnicity | Race × Phase | Same as GLM-1 |
| GLM-1c | Race/Ethnicity | Race × Model Type | Same as GLM-1 |
| GLM-1d | Race/Ethnicity | Race × Mobility Used | Same as GLM-1 |
| *Main + Interaction Effects: Urbanicity Interaction Models* | | | |
| GLM-2a | Urbanicity | Urbanicity × Lookahead | Same as GLM-2 |
| GLM-2b | Urbanicity | Urbanicity × Phase | Same as GLM-2 |
| GLM-2c | Urbanicity | Urbanicity × Model Type | Same as GLM-2 |
| GLM-2d | Urbanicity | Urbanicity × Mobility Used | Same as GLM-2 |

**Distribution Family:** Gaussian with log link function for all models
**Health Outcomes:** Asthma, Obesity, COPD, CHD, CKD, Diabetes, Obesity, Cancer, Stroke, Age 65+
**Data-Model Characteristics:** Lookaheads (7(reference), 14, 21, 28), Phase (0 (reference) 1-6), Model Type (Compartmental (reference), Baseline, Deep Learning, Ensemble, Statistical), Mobility Usage (No (reference), Yes, Mixed)
**Dependent Variable:** Square root of PBL (sqrt_pbl) with 1% trimming
**Model Assumptions:** Independent observations, exponential family distribution (Gaussian), linear predictor through log link function, constant variance of residuals on link scale, no perfect multicollinearity (adjusted GVIF < 2)

**Table 3. Summary of regression results: GLM-1.**

| Variable | e$^{Coef.}$ (Std. Error) | 95% CI | z-value |
|---|---|---|---|
| Intercept | 0.009*** (0.009) | [0.009, 0.009] | -510.099 |
| | *Sensitive Attributes* | | |
| % Black | 0.976*** (0.003) | [0.970, 0.983] | −6.864 |
| % Hispanic | 1.216*** (0.003) | [1.209, 1.223] | 63.844 |
| % Asian | 0.515*** (0.015) | [0.500, 0.530] | −44.647 |
| | *Lookahead Period* | | |
| 14 days | 1.119*** (0.001) | [1.118, 1.121] | 143.586 |
| 21 days | 1.211*** (0.001) | [1.209, 1.213] | 249.396 |
| 28 days | 1.300*** (0.001) | [1.298, 1.302] | 352.668 |
| | *Phase Effects* | | |
| Phase 1 | 1.445*** (0.001) | [1.443, 1.448] | 377.204 |
| Phase 2 | 0.890*** (0.001) | [0.888, 0.892] | −99.535 |
| Phase 3 | 1.285*** (0.001) | [1.283, 1.288] | 236.425 |
| Phase 4 | 1.553*** (0.001) | [1.549, 1.556] | 416.701 |
| Phase 5 | 0.969*** (0.001) | [0.967, 0.972] | −22.748 |
| Phase 6 | 1.152*** (0.001) | [1.149, 1.154] | 112.373 |
| | *Model Type* | | |
| Baseline | 1.121*** (0.002) | [1.118, 1.124] | 72.937 |
| Deep Learning | 1.037*** (0.001) | [1.035, 1.039] | 40.193 |
| Ensemble | 1.007*** (0.001) | [1.005, 1.009] | 7.959 |
| Statistical | 1.076*** (0.001) | [1.074, 1.078] | 89.646 |
| | *Mobility* | | |
| Mixed | 0.864*** (0.001) | [0.862, 0.866] | −124.255 |
| Yes | 0.996*** (0.001) | [0.994, 0.997] | −5.637 |
| | *Health Controls* | | |
| Asthma | 0.959*** (0.001) | [0.957, 0.961] | −43.776 |
| Obesity | 1.003*** (0.000) | [1.003, 1.004] | 25.942 |
| COPD | 1.046*** (0.000) | [1.045, 1.047] | 93.094 |
| % Age 65+ | 1.214*** (0.007) | [1.198, 1.230] | 28.712 |

**Model Statistics:** Pseudo $R^2$ (Cox-Snell) = 0.460; Log-Likelihood = 5,981,789; N = 1,526,869
**Notes:** ***$p < 0.001$, **$p < 0.01$, *$p < 0.05$. *Dependent Variable:* sqrt_pbl. *Link Function:* log. *Regression Family:* Gaussian. State fixed effects included but are reported in Table 5 in S1 Appendix. *exp(Coef)* represents the multiplicative effect on the outcome. CI: Confidence Interval.

lower prediction errors than the White population baseline ($\beta$ = −0.663, exp($\beta$) = 0.515, $p < 0.001$). Our results indicate that for every 1% increase in the Asian population (with respect to White population), the prediction accuracy is approximately 48.5% better when compared to the White group. For predominantly Black communities, we observed a slight but statistically significant improvement in prediction accuracy ($\beta$ = −0.024, exp($\beta$) = 0.976, $p < 0.001$), with 1% increase in Black population (with respect to White) being associated with predictions approximately 2.4% more accurate than the baseline.

**Model diagnostics.** Model assessments indicate that Gaussian-GLM demonstrates adequate fit and reliability. The pseudo-$R^2$ is 0.46, and the residuals show an approximately normal distribution with mild heteroskedasticity at extreme values. Cook's distance analysis identified no influential points that would substantially affect our findings. Although there are some deviations from normality in the tails of the residual distribution, these do not materially affect our main conclusions regarding demographic disparities in prediction accuracy (detailed diagnostics are provided in Sect 2.3.1, Fig 3 in S1 Appendix).

**Summary**: Race

COVID-19 forecast errors show substantial racial disparities, particularly for **Hispanic** Race:

⇒ 21.6% *higher* for Hispanic populations
⇒ 48.5% *lower* for Asian populations
⇒ 2.4% *lower* for Black population

## Fairness of COVID-19 case predictions across urbanization levels (GLM-2)

Table 4 illustrates the estimated coefficients of the generalized linear model for the fairness analysis of COVID-19 prediction accuracy across urbanization levels (GLM-2, Eq 2). The table reveals insights into the relationship between urbanization levels and the PBL while controlling for the impact of various data-model features, health outcomes and state fixed effects on prediction accuracy.

Relative to LM areas, prediction errors varied significantly across urbanization levels. MC areas, which are urban clusters with populations up to 50,000, showed notably higher prediction errors ($\beta = 0.063$, $\exp(\beta) = 1.065$, $p < 0.001$), indicating approximately 6.5% higher errors in these regions. This suggests particular challenges in forecasting COVID-19 cases in smaller urban areas that may have different healthcare infrastructure and reporting systems compared to LM areas. SMM areas also demonstrated increased prediction errors ($\beta = 0.027$, $\exp(\beta) = 1.027$, $p < 0.001$), though the effect was more modest with 2.7% higher errors than large metropolitan regions.

**Model diagnostics.** Our model assessments reveal that GLM-2 achieves similar statistical properties to GLM-1, with a pseudo-$R^2$ of 0.46. While the residual distribution shows some deviation from normality at the tails, these departures do not substantially impact our key findings regarding urbanicity-based disparities in prediction accuracy. The complete characteristics of the model and the GVIF analysis can be found in Sect 2.3.2, Fig 4 in S1 Appendix.

**Summary**: Urbanicity

⇒ Prediction disparities worsen for more rural areas.

## Fairness of COVID-19 case predictions across race, ethnicity and model-data characteristics

**GLM-1a: Race, ethnicity and forecast lookahead.** Table 5 presents the interaction effects between race, ethnicity and lookahead periods, revealing notable variations in forecast accuracy across different time horizons, and pointing to unequal (unfair) error distributions (see Eq 3). The trends described for the main effects (GLM-1) persist when adding the interaction effects *i.e.,* increases in Black or Asian population in a county (with respect to White) and for a given lookahead, are associated with lower PBLs; while increases in Hispanic population are associated with higher PBLs.

Our analysis reveals that increases in Hispanic population (with respect to White) are related to persistent disparities in forecast accuracy across all prediction horizons, though these disparities show a gradual decrease for predictions over longer timeframes. When examining the relative effects, we find that areas with higher Hispanic populations experience prediction errors that are 24.6% higher than predominantly White areas at 14-day forecasts,

**Table 4. Summary of regression results: GLM-2.**

| Variable | $e^{Coef.}$ (Std. Error) | 95% CI | z-value |
|---|---|---|---|
| Intercept | 0.009*** (0.018) | [0.009, 0.009] | -263.017 |
| | *Sensitive Attributes* | | |
| Micropolitan | 1.065*** (0.001) | [1.063, 1.067] | 60.326 |
| Small and Medium Metro | 1.027*** (0.001) | [1.025, 1.029] | 26.509 |
| | *Lookahead Period* | | |
| 14 days | 1.119*** (0.001) | [1.117, 1.121] | 143.691 |
| 21 days | 1.211*** (0.001) | [1.209, 1.213] | 249.533 |
| 28 days | 1.299*** (0.001) | [1.298, 1.301] | 352.841 |
| | *Phase Effects* | | |
| Phase 1 | 1.445*** (0.001) | [1.442, 1.448] | 377.540 |
| Phase 2 | 0.890*** (0.001) | [0.888, 0.892] | −100.050 |
| Phase 3 | 1.285*** (0.001) | [1.282, 1.287] | 236.167 |
| Phase 4 | 1.552*** (0.001) | [1.549, 1.555] | 416.754 |
| Phase 5 | 0.969*** (0.001) | [0.967, 0.972] | −22.827 |
| Phase 6 | 1.151*** (0.001) | [1.148, 1.154] | 112.311 |
| | *Model Type* | | |
| Baseline | 1.121*** (0.002) | [1.118, 1.124] | 73.043 |
| Deep Learning | 1.037*** (0.001) | [1.035, 1.039] | 40.217 |
| Ensemble | 1.007*** (0.001) | [1.005, 1.009] | 7.952 |
| Statistical | 1.076*** (0.001) | [1.074, 1.078] | 89.919 |
| | *Mobility* | | |
| Mixed | 0.864*** (0.001) | [0.862, 0.866] | −124.509 |
| Yes | 0.996*** (0.001) | [0.994, 0.997] | −5.631 |
| | *Health Controls* | | |
| Asthma | 0.956*** (0.001) | [0.954, 0.958] | −34.853 |
| Obesity | 1.004*** (0.000) | [1.004, 1.005] | 32.018 |
| BPHigh | 0.998*** (0.000) | [0.997, 0.998] | −7.667 |
| COPD | 1.000 (0.001) | [0.998, 1.002] | −0.089 |
| Stroke | 0.981*** (0.003) | [0.974, 0.988] | −5.605 |
| Cancer | 0.984*** (0.003) | [0.979, 0.990] | −5.650 |
| CHD | 1.089*** (0.003) | [1.084, 1.095] | 33.418 |
| Diabetes | 0.995*** (0.001) | [0.993, 0.997] | −5.442 |
| CKD | 1.043*** (0.006) | [1.030, 1.056] | 6.559 |
| % Age 65+ | 1.111*** (0.007) | [1.096, 1.126] | 15.328 |

**Model Statistics:** Pseudo $R^2$ (CS) = 0.462; Log-Likelihood = 5,983,292; N = 1,526,869
**Notes:** ***$p < 0.001$, **$p < 0.01$, *$p < 0.05$. *Dependent Variable:* sqrt_pbl. *Link Function:* log. *Regression Family:* Gaussian. State fixed effects included but are reported in Table 6 in S1 Appendix. $exp(Coef)$ represents the multiplicative effect on the outcome. CI: Confidence Interval.

declining to 18.5% at 21-day forecasts, and further reducing to 16.8% at 28-day forecasts. This pattern indicates that although the disparity in prediction accuracy moderates somewhat over longer forecast horizons, significant inequities in model performance persist throughout all prediction timeframes for Hispanic communities.

For Black counties, the relative effect shows progressively decreasing significant differences in forecast errors between Black and the White reference group, with PBL errors being 3.1% higher for White counties for the 14–day forecast ($exp(\beta)$ = 0.969, $p < 0.001$), decreasing to 1.3% higher for 21 days ($exp(\beta)$ = 0.987, $p < 0.001$). By the 28-day forecast horizon, the prediction accuracy for Black counties shows no significant difference from White areas (relative effect: 1.000, 0.0% difference).

Asian counties reveal the most pronounced variation across forecast horizons. The relative effect analysis shows that while predictions remain more accurate than for White counties across all horizons, this advantage decreases substantially over longer forecast periods. The

**Table 5. GLM-1a: Race × lookahead effects relative to white reference group.**

| Variable | Coefficient Estimates | | Relative Effect | |
|---|---|---|---|---|
| | $e^{\text{Coef.}}$ (SE) | 95% CI | $e^{\text{Coef.}}$ | % Diff from White |
| % Asian (7-day ref.) | 0.228*** (0.030) | [0.215, 0.243] | 0.228*** | −77.2% |
| × 14-day ahead | 1.652*** (0.037) | [1.536, 1.777] | 0.377*** | −62.3% |
| × 21-day ahead | 2.689*** (0.035) | [2.509, 2.882] | 0.613*** | −38.7% |
| × 28-day ahead | 3.505*** (0.034) | [3.279, 3.746] | 0.799*** | −20.1% |
| % Black (7-day ref.) | 0.937*** (0.005) | [0.928, 0.947] | 0.937*** | −6.3% |
| × 14-day ahead | 1.034*** (0.006) | [1.023, 1.046] | 0.969*** | −3.1% |
| × 21-day ahead | 1.053*** (0.005) | [1.042, 1.065] | 0.987*** | −1.3% |
| × 28-day ahead | 1.067*** (0.005) | [1.056, 1.078] | 1.000 | 0.0% |
| % Hispanic (7-day ref.) | 1.298*** (0.005) | [1.287, 1.310] | 1.298*** | +29.8% |
| × 14-day ahead | 0.960*** (0.005) | [0.951, 0.970] | 1.246*** | +24.6% |
| × 21-day ahead | 0.913*** (0.005) | [0.904, 0.923] | 1.185*** | +18.5% |
| × 28-day ahead | 0.900*** (0.005) | [0.891, 0.909] | 1.168*** | +16.8% |

**Model Statistics:** Pseudo $R^2$ (CS) = 0.461; Log-Likelihood = 5,982,892; N = 1,526,869

**Notes:** ***$p < 0.001$, **$p < 0.01$, *$p < 0.05$. *Dependent Variable:* Square root PBL. *Link Function:* Log. *Regression Family:* Gaussian. The table shows the GLM coefficients and their significance for model GLM-1a. We only discuss race, ethnicity and lookahead interaction coefficients. For clarity purposes, all other main effects and control variables: Health outcomes, age 65+ and state fixed effects are only shown and discussed in Tables 7, 8 in S1 Appendix. Model Diagnostics are provided in Fig 5a in S1 Appendix. For the Coefficient Estimates, *exp(Coef)* represents the multiplicative effect on the outcome, SE: Standard Error and CI: Confidence Interval. The Relative Effect represents the multiplicative effect on the forecast error (PBL) of a particular race or ethnicity compared to the White population within each lookahead. The relative effect is represented by *exp(Coef)* and computed as $e^{\beta_i + \delta_{ij}}$ with coefficients from Eq 3. To evaluate better the relative effect, we also discuss the percentage change in forecast error when compared to White population for each lookahead variable (*% Diff from White*). This change is computed as $(1 - e^{\beta_i + \delta_{ij}}) * 100\%$ for a given race/ethnicity and lookahead value, and it represents the percentage increase or decrease in the forecast error (PBL) with respect to the White population (e.g., +24.6% means that the PBL error for Hispanic counties at 14-day lookahead is 24.6% higher PBL when compared to White). All effects should be interpreted as the relative difference compared to White population within each specific lookahead value. The relative coefficient significance is evaluated using `linearHypothesis` in R (*car* package [59])

relative effect shows that areas with higher Asian populations have prediction errors 62.3% lower than White areas at 14-day forecasts, improving to 38.7% lower at 21-day forecasts, and 20.1% lower at 28-day forecasts.

**GLM-1b: Race, ethnicity and COVID-19 phase.** When examining how racial and ethnic disparities vary across different pandemic phases, we observe substantial heterogeneity in PBL, with patterns varying markedly by both race and phase (see Table 6). The results reveal complex temporal dynamics in prediction fairness across different demographic groups.

For Hispanic populations, the analysis reveals significant variation in prediction errors across pandemic phases relative to White populations. At initial stages of the pandemic (Phase 0), Hispanic counties show substantially higher errors (+101.8%) compared to White counties. While this disparity persists across most phases, its magnitude fluctuates notably. Phase 1 also exhibits a higher disparity (+36.6%), followed by Phase 3 (+15.6%) and Phase 6 (+12.0%). However, in Phases 4 and 5, this pattern reverses, with Hispanic counties showing slightly lower errors than White counties (−8.2% and −0.3% respectively), suggesting that prediction fairness for Hispanic communities varied significantly with pandemic phase characteristics.

For Black populations, the results show complex phase-dependent patterns. Starting with notably higher errors in Phase 0 (+113.0% compared to White areas), the disparities shift dramatically across phases. Phase 4 shows the most favorable performance for Black communities, with errors 42.5% lower than White areas. However, Phases 3 and 6 show notably higher

**Table 6. GLM-1b: Race × phase effects relative to white reference group.**

| Variable | Coefficient Estimates $e^{Coef.}$ (SE) | 95% CI | Relative Effect $e^{Coef.}$ | % Diff from White |
|---|---|---|---|---|
| % Asian (Phase 0 ref.) | 0.137*** (0.042) | [0.126, 0.149] | 0.137*** | −86.3% |
| × Phase 1 | 2.092*** (0.046) | [1.912, 2.290] | 0.287*** | −71.3% |
| × Phase 2 | 8.767*** (0.049) | [7.957, 9.659] | 1.201*** | +20.1% |
| × Phase 3 | 0.812*** (0.053) | [0.732, 0.900] | 0.111*** | −88.9% |
| × Phase 4 | 6.310*** (0.047) | [5.756, 6.918] | 0.864*** | −13.6% |
| × Phase 5 | 35.577*** (0.049) | [32.323, 39.159] | 4.871*** | +387.1% |
| × Phase 6 | 3.260*** (0.057) | [2.916, 3.645] | 0.447*** | −55.3% |
| % Black (Phase 0 ref.) | 2.130*** (0.005) | [2.108, 2.153] | 2.130*** | 113.0% |
| × Phase 1 | 0.438*** (0.006) | [0.434, 0.443] | 0.933*** | −6.7% |
| × Phase 2 | 0.393*** (0.007) | [0.388, 0.399] | 0.837*** | −16.3% |
| × Phase 3 | 0.541*** (0.006) | [0.535, 0.548] | 1.152*** | +15.2% |
| × Phase 4 | 0.270*** (0.007) | [0.267, 0.274] | 0.575*** | −42.5% |
| × Phase 5 | 0.375*** (0.009) | [0.369, 0.382] | 0.799*** | −20.1% |
| × Phase 6 | 0.529*** (0.008) | [0.521, 0.536] | 1.127*** | +12.7% |
| % Hispanic (Phase 0 ref.) | 2.018*** (0.005) | [1.998, 2.039] | 2.018*** | 101.8% |
| × Phase 1 | 0.677*** (0.006) | [0.670, 0.685] | 1.366*** | +36.6% |
| × Phase 2 | 0.549*** (0.007) | [0.541, 0.557] | 1.108*** | +10.8% |
| × Phase 3 | 0.573*** (0.006) | [0.566, 0.580] | 1.156*** | +15.6% |
| × Phase 4 | 0.455*** (0.006) | [0.449, 0.460] | 0.918*** | −8.2% |
| × Phase 5 | 0.494*** (0.009) | [0.485, 0.503] | 0.997 | −0.3% |
| × Phase 6 | 0.555*** (0.008) | [0.547, 0.564] | 1.120*** | +12.0% |

**Model Statistics:** Pseudo $R^2$ (CS) = 0.495; Log-Likelihood = 6,012,390; N = 1,526,869

**Notes:** ***$p < 0.001$, **$p < 0.01$, *$p < 0.05$. *Dependent Variable:* Square root PBL. *Link Function:* Log. *Regression Family:* Gaussian. The table shows the GLM coefficients and their significance for model GLM-1b. We only discuss race, ethnicity and phase interaction coefficients. For clarity purposes, all other main effects and control variables: Health outcomes, age 65+ and state fixed effects are only shown and discussed in Tables 9, 10 in S1 Appendix. Model Diagnostics are provided in Fig 5b in S1 Appendix. For the Coefficient Estimates, *exp(Coef)* represents the multiplicative effect on the outcome, SE: Standard Error and CI: Confidence Interval. The Relative Effect represents the multiplicative effect on the forecast error (PBL) of a particular race or ethnicity compared to the White population within each phase. The relative effect is represented by *exp(Coef)* and computed as $e^{\beta_i + \delta_{ij}}$ with coefficients from Eq 3. To evaluate better the relative effect, we also discuss the percentage change in forecast error when compared to White population for each phase variable (*% Diff from White*). This change is computed as $(1 - e^{\beta_i + \delta_{ij}}) * 100\%$ for a given race/ethnicity and phase value, and it represents the percentage increase or decrease in the forecast error (PBL) with respect to the White population (e.g., +101.8% means that the PBL error for Hispanic counties in Phase 0 is 101.8% higher PBL when compared to White). All effects should be interpreted as the relative difference compared to White population within each specific phase value. The relative coefficient significance is evaluated using `linearHypothesis` in R (*car* package [59])

prediction errors (+15.2% and +12.7% respectively) compared to White areas. Phases 1, 2, and 5 show better performance for Black communities (-6.7%, -16.3%, and -20.1% respectively).

Asian populations exhibit the most extreme phase-dependent variations in prediction accuracy. While starting with substantially lower errors than White areas in Phase 0 (-86.3%), the disparities show dramatic swings across phases. Phase 5 stands out with strikingly higher errors (+387.1% compared to White areas), while Phase 3 shows the best relative performance (-88.9%). Phase 2 also shows notably higher errors (+20.1%), while Phases 1, 4, and 6 maintain lower errors compared to White areas (-71.3%, -13.6%, and -55.3% respectively). These extreme variations suggest that predictions for Asian communities were particularly sensitive to phase-specific characteristics of the pandemic.

All these phase-dependent variations are statistically significant ($p < 0.001$), and they reveal that prediction fairness across racial and ethnic groups is not consistent throughout the pandemic, with certain phases (particularly Phase 0) associated with the largest disparities relative to White populations.

**GLM-1c: Race, ethnicity and model type.** Our analysis of how prediction disparities vary across different minority race/ethnic groups and model types with respect to the White reference group reveals distinct patterns and demonstrates that model architecture choices have significant implications for prediction fairness (see Table 7).

Increases in Hispanic population with respect to the White reference group are associated with large performance disparities across all model types. Baseline models show the largest disparity (28.1% higher PBL compared to White areas), while Compartmental models show somewhat reduced, though still substantial, disparities (+30.7%). Ensemble models register the lowest disparity (+18.3%).

Increases in Black population are associated with relatively modest variations in PBL across model types when compared to the White reference group. Compartmental and Baseline Models show marginally higher prediction errors (0.9% & +2.1% difference with White areas), while Deep Learning & Ensemble models demonstrate better relative performance (-9.0%, -5%). Statistical models perform better for Black communities (-4.1%).

The Asian subgroup, on the other hand, shows improved relative effect with respect to White populations, and these disparities remain relatively consistent across model types, with PBL errors being at least 24.3% lower than errors for the White group across all model types.

**Table 7. GLM-1c: Race × model type effects relative to white reference group.**

| Variable | Coefficient Estimates | | Relative Effect | |
|---|---|---|---|---|
| | $e^{Coef.}$ (SE) | 95% CI | $e^{Coef.}$ | % Diff from White |
| % Asian (Compartmental ref.) | 0.463*** (0.019) | [0.446, 0.481] | 0.463*** | −53.7% |
| × Baseline Models | 1.635*** (0.046) | [1.494, 1.789] | 0.757*** | −24.3% |
| × Deep Learning | 1.549*** (0.042) | [1.427, 1.681] | 0.717*** | −28.3% |
| × Ensemble | 1.232*** (0.027) | [1.168, 1.300] | 0.571*** | −42.9% |
| × Statistical | 0.934* (0.033) | [0.876, 0.995] | 0.432*** | −56.8% |
| % Black (Compartmental ref.) | 1.009* (0.004) | [1.001, 1.017] | 1.009* | +0.9% |
| × Baseline Models | 1.012 (0.008) | [0.995, 1.028] | 1.021* | +2.1% |
| × Deep Learning | 0.902*** (0.006) | [0.890, 0.913] | 0.910*** | −9.0% |
| × Ensemble | 0.942*** (0.005) | [0.933, 0.950] | 0.950*** | −5.0% |
| × Statistical | 0.951*** (0.005) | [0.941, 0.960] | 0.959*** | −4.1% |
| % Hispanic (Compartmental ref.) | 1.235*** (0.004) | [1.226, 1.243] | 1.235*** | +23.5% |
| × Baseline Models | 1.037*** (0.008) | [1.021, 1.053] | 1.281*** | +28.1% |
| × Deep Learning | 0.966*** (0.006) | [0.955, 0.978] | 1.193*** | +19.3% |
| × Ensemble | 0.958*** (0.004) | [0.950, 0.967] | 1.183*** | +18.3% |
| × Statistical | 0.982*** (0.005) | [0.972, 0.991] | 1.213*** | +21.3% |

**Model Statistics:** Pseudo $R^2$ (CS) = 0.461; Log-Likelihood = 5,982,160; N = 1,526,869

**Notes:** ***$p < 0.001$, **$p < 0.01$, *$p < 0.05$. *Dependent Variable:* Square root PBL. *Link Function:* Log. *Regression Family:* Gaussian. The table shows the GLM coefficients and their significance for model GLM-1c. We only discuss race, ethnicity and model type interaction coefficients. For clarity purposes, all other main effects and control variables: Health outcomes, age 65+ and state fixed effects are only shown and discussed in Tables 11, 12 in S1 Appendix. Model Diagnostics are provided in Fig 5c in S1 Appendix. For the Coefficient Estimates, *exp(Coef)* represents the multiplicative effect on the outcome, SE: Standard Error and CI: Confidence Interval. The Relative Effect represents the multiplicative effect on the forecast error (PBL) of a particular race or ethnicity compared to the White population within each model type. The relative effect is represented by *exp(Coef)* and computed as $e^{\beta_i + \delta_{ij}}$ with coefficients from Eq 3. To evaluate better the relative effect, we also discuss the percentage change in forecast error when compared to White population for each model type (*% Diff from White*). This change is computed as $(1 - e^{\beta_i + \delta_{ij}}) * 100\%$, for a given race/ethnicity and model type, and it represents the percentage increase or decrease in the forecast error (PBL) with respect to the White population (e.g., +23.5% means that the PBL error for Hispanic counties for Compartmental models is 23.5% higher PBL when compared to White). All effects should be interpreted as the relative difference compared to White population within each specific model type. The relative coefficient significance is evaluated using `linearHypothesis` in R (*car* package [59])

These findings suggest that model architecture choices significantly impact prediction fairness, with Deep Learning and Ensemble models providing the most balanced performance across racial and ethnic groups, particularly for Hispanic and Black populations

**GLM-1d: Race, ethnicity and mobility data.** Analysis of how prediction disparities vary with mobility data usage also reveal interesting patterns in the relationship between data inputs and prediction fairness (Table 8).

The relative effect analysis for Hispanic population indicates significant disparities in forecast performance when compared to the White reference group across mobility data uses in prediction models. For forecast models that use mobility data, increases in Hispanic population with respect to the White reference group are associated with a PBL 18.5% higher than that of the White population. That number increases to 22% for mixed models (ensembles of CDC ForecastHub models trained with and without mobility data). On the other hand, when mobility data is not used, PBL errors are 31.2% higher for each 1% increase in Hispanic population with respect to the White reference group, suggesting that mobility data usage can improve prediction fairness for Hispanic communities. However, considerable disparities remain (higher PBL errors) when compared to the White population.

For Black population, and when compared to the White reference group, not using mobility data in the forecast models was associated with PBL errors being 6.8% higher. However, using mobility data reversed that trend, with PBL errors being 5.1% lower when the Black population increases 1% with respect to the White group.

Prediction performance for Asian populations when considering the use of mobility data was significantly better than the performance for White counties, independently of whether mobility data was used or not in the models, with PBL errors being between 42.9% and 64.9% significantly lower than errors for the White reference group. Interestingly, we also observe

**Table 8. GLM-1d: Race × mobility data inclusion effects relative to white.**

| Variable | Coefficient Estimates | | Relative Effect | |
|---|---|---|---|---|
| | $e^{Coef.}$ (SE) | 95% CI | $e^{Coef.}$ | % Diff from White |
| % Asian (No Mobility ref.) | 0.351*** (0.027) | [0.332, 0.370] | 0.351*** | −64.9% |
| × Mixed | 1.585*** (0.035) | [1.480, 1.697] | 0.556*** | −44.4% |
| × Mobility Used | 1.626*** (0.029) | [1.536, 1.722] | 0.571*** | −42.9% |
| % Black (No Mobility ref.) | 1.068*** (0.005) | [1.058, 1.079] | 1.068*** | +6.8% |
| × Mixed | 0.909*** (0.006) | [0.899, 0.919] | 0.971*** | −2.9% |
| × Mobility Used | 0.889*** (0.005) | [0.881, 0.897] | 0.949*** | −5.1% |
| % Hispanic (No Mobility ref.) | 1.312*** (0.005) | [1.301, 1.324] | 1.312*** | +31.2% |
| × Mixed | 0.930*** (0.005) | [0.920, 0.940] | 1.220*** | +22.0% |
| × Mobility Used | 0.903*** (0.004) | [0.895, 0.911] | 1.185*** | +18.5% |

**Model Statistics:** Pseudo $R^2$ (CS) = 0.461; Log-Likelihood = 5,982,399; N = 1,526,869

**Notes:** ***$p < 0.001$, **$p < 0.01$, *$p < 0.05$. *Dependent Variable:* Square root PBL. *Link Function:* Log. *Regression Family:* Gaussian. The table shows the GLM coefficients and their significance for model GLM-1d. We only discuss race, ethnicity and mobility usage interaction coefficients. For clarity purposes, all other main effects and control variables: Health outcomes, age 65+ and state fixed effects are only shown and discussed Tables 13, 14 in S1 Appendix. Model Diagnostics are provided in Fig 5d in S1 Appendix. For the Coefficient Estimates, *exp(Coef)* represents the multiplicative effect on the outcome, SE: Standard Error and CI: Confidence Interval. The Relative Effect represents the multiplicative effect on the forecast error (PBL) of a particular race or ethnicity compared to the White population within each mobility usage category. The relative effect is represented by *exp(Coef)* and computed as $e^{\beta_i + \delta_{ij}}$ with coefficients from Eq 3. To evaluate better the relative effect, we also discuss the percentage change in forecast error when compared to White population for each mobility usage category (*% Diff from White*). This change is computed as $(1 - e^{\beta_i + \delta_{ij}}) * 100\%$, for a given race/ethnicity and mobility usage category, and it represents the percentage increase or decrease in the forecast error (PBL) with respect to the White population (e.g., +31.2% means that the PBL error for Hispanic counties for the given mobility usage category is 31.2% higher PBL when compared to White). All effects should be interpreted as the relative difference compared to White population within each specific mobility usage category. The relative coefficient significance is evaluated using `linearHypothesis` in R (*car* package [59])

that use of mobility data helps reduce the disparity in performance errors (42.9% and 44.4% vs. 64.9%).

These results reveal significant prediction performance differences between minority racial and ethnic groups with respect to White across mobility data use settings; while suggesting that the incorporation of mobility data might help reduce prediction disparities for minority racial groups.

---

**Summary**: Race X Model-Data Characteristics

⇒ Hispanic communities consistently experience higher prediction errors compared to White areas across most configurations.
⇒ Disparities decrease with longer forecast horizons.
⇒ Initial phases of the pandemic saw widest disparity for both Hispanic and Black subgroups
⇒ Asian communities maintained lower error rates overall & showed substantial phase-dependent variations (-88.9% to +387.1%)
⇒ Deep Learning and Ensemble models demonstrated most balanced performance across all racial and ethnic groups
⇒ Mobility data usage generally helps reduce prediction disparities

---

## Fairness of COVID-19 case predictions across urbanicity and model-data characteristics

**GLM-2a: Urbanicity and forecast lookahead.** Table 9 exhibits significant disparities in COVID-19 case prediction errors between urbanization levels across different forecast horizons. Our findings demonstrate that both MC and SMM areas consistently experience higher

**Table 9. GLM-2a: Urbanicity × lookahead effects relative to LM reference group.**

| Variable | Coefficient Estimates | | Relative Effect | |
|---|---|---|---|---|
| | $e^{Coef.}$ (SE) | 95% CI | $e^{Coef.}$ | % Diff from LM |
| **Micropolitan (MC)** (7-day ref.) | 1.133*** (0.002) | [1.128, 1.137] | 1.133*** | +13.3% |
| × 14-day ahead | 0.963*** (0.003) | [0.958, 0.968] | 1.091*** | +9.1% |
| × 21-day ahead | 0.926*** (0.003) | [0.921, 0.930] | 1.049*** | +4.9% |
| × 28-day ahead | 0.900*** (0.002) | [0.896, 0.905] | 1.020*** | +2.0% |
| **Small/Medium Metro (SMM)** (7-day ref.) | 1.048*** (0.002) | [1.044, 1.053] | 1.048*** | +4.8% |
| × 14-day ahead | 0.984*** (0.003) | [0.979, 0.990] | 1.031*** | +3.1% |
| × 21-day ahead | 0.975*** (0.003) | [0.970, 0.981] | 1.022*** | +2.2% |
| × 28-day ahead | 0.969*** (0.003) | [0.964, 0.975] | 1.015*** | +1.5% |

**Model Statistics:** Pseudo $R^2$ (CS) = 0.464; Log-Likelihood = 5,985,040; N = 1,526,869

**Notes:** ***$p < 0.001$, **$p < 0.01$, *$p < 0.05$. *Dependent Variable:* Square root PBL. *Link Function:* Log. *Regression Family:* Gaussian. The table shows the GLM coefficients and their significance for model GLM-2a. We only discuss urbanicity and lookahead interaction coefficients. For clarity purposes, all other main effects and control variables: Health outcomes, age 65+ and state fixed effects are only shown and discussed in Tables 15, 16 in S1 Appendix. Model Diagnostics are provided in Fig 6a in S1 Appendix. For the Coefficient Estimates, *exp(Coef)* represents the multiplicative effect on the outcome, SE: Standard Error and CI: Confidence Interval. The Relative Effect represents the multiplicative effect on the forecast error (PBL) of a particular urbanicity level compared to the Large Metropolitan (LM) areas within each lookahead. The relative effect is represented by *exp(Coef)* and computed as $e^{\beta_i + \delta_{ij}}$ with coefficients from Eq 4. To evaluate better the relative effect, we also discuss the percentage change in forecast error when compared to LM areas for each lookahead variable (*% Diff from LM*). This change is computed as $(1 - e^{\beta_i + \delta_{ij}}) * 100\%$ for a given urbanicity level and lookahead value, and it represents the percentage increase or decrease in the forecast error (PBL) with respect to LM areas. All effects should be interpreted as the relative difference compared to LM areas within each specific lookahead value. The relative coefficient significance is evaluated using `linearHypothesis` in R (*car* package [59])

prediction errors when compared to LM areas (reference group) across lookaheads; and that the magnitude of these disparities decreases as the prediction horizon extends.

MC areas show the most pronounced disparity, with baseline (7-day) prediction errors 13.3% higher than LM areas. This disparity, while persistent, diminishes at longer prediction horizons. For 14-day forecasts, the relative effect shows that MC areas still experience 9.1% higher errors than LM areas. This gap continues to narrow, decreasing to 4.9% for 21-day forecasts and further reducing to 2.0% for 28-day predictions.

SMM areas exhibit a similar pattern but with smaller magnitudes of disparity compared to MC. These areas show prediction errors 4.8% higher than LM areas for 7-day forecasts. Again the pattern of the disparity decreasing is also observed for SMMs, with relative effects showing 3.1% higher errors for 14-day forecasts, 2.2% for 21-day forecasts, and 1.5% for 28-day forecasts compared to LM areas. These differences, while smaller, remain statistically significant across all forecast horizons ($p < 0.001$).

**GLM-2b: Urbanicity and COVID-19 phase.** Both MC and SMM areas show significantly different prediction performance compared to LM areas, with these disparities varying substantially across different pandemic phases (See Table 10).

MC areas demonstrate the highest baseline disparity (Phase 0), with prediction errors 11.6% higher than LM areas. The magnitude of this disparity fluctuates notably across different pandemic phases. During Phase 3, MC areas experienced their largest disparity, with errors 12.4% higher than LM areas. Conversely, Phases 2 and 5 show a reversal of this pattern, with MC areas actually performing better than LM areas, showing 4.2% and 8.5% lower errors

**Table 10. GLM-2b: Urbanicity × phase effects relative to LM areas.**

| Variable | Coefficient Estimates | | Relative Effect | |
|---|---|---|---|---|
| | $e^{Coef.}$ (SE) | 95% CI | $e^{Coef.}$ | % Diff from LM |
| **Micropolitan (MC)** (Phase 0 ref.) | 1.116*** (0.003) | [1.110, 1.122] | 1.116*** | +11.6% |
| × Phase 1 | 0.997 (0.003) | [0.991, 1.003] | 1.113*** | +11.3% |
| × Phase 2 | 0.858*** (0.004) | [0.852, 0.864] | 0.958*** | −4.2% |
| × Phase 3 | 1.007* (0.003) | [1.001, 1.014] | 1.124*** | +12.4% |
| × Phase 4 | 0.915*** (0.003) | [0.909, 0.921] | 1.021*** | +2.1% |
| × Phase 5 | 0.820*** (0.004) | [0.814, 0.827] | 0.915*** | −8.5% |
| × Phase 6 | 1.004 (0.004) | [0.996, 1.013] | 1.120*** | +12.0% |
| **Small/Medium Metro (SMM)** (Phase 0 ref.) | 1.091*** (0.003) | [1.085, 1.098] | 1.091*** | +9.1% |
| × Phase 1 | 0.960*** (0.004) | [0.953, 0.966] | 1.047*** | +4.7% |
| × Phase 2 | 0.876*** (0.004) | [0.869, 0.883] | 0.956*** | −4.4% |
| × Phase 3 | 0.998 (0.004) | [0.991, 1.006] | 1.089*** | +8.9% |
| × Phase 4 | 0.898*** (0.004) | [0.891, 0.904] | 0.980*** | −2.0% |
| × Phase 5 | 0.862*** (0.005) | [0.854, 0.870] | 0.940*** | −6.0% |
| × Phase 6 | 0.971*** (0.005) | [0.963, 0.980] | 1.059*** | +5.9% |

**Model Statistics:** Pseudo $R^2$ (CS) = 0.466; Log-Likelihood = 5,987,062; N = 1,526,869

**Notes:** ***$p < 0.001$, **$p < 0.01$, *$p < 0.05$. *Dependent Variable:* Square root PBL. *Link Function:* Log. *Regression Family:* Gaussian. The table shows the GLM coefficients and their significance for model GLM-2b. We only discuss urbanicity and phase interaction coefficients. For clarity purposes, all other main effects and control variables: Health outcomes, age 65+ and state fixed effects are only shown and discussed in Tables 17, 18 in S1 Appendix. Model Diagnostics are provided in Fig 6b in S1 Appendix. For the Coefficient Estimates, *exp(Coef)* represents the multiplicative effect on the outcome, SE: Standard Error and CI: Confidence Interval. The Relative Effect represents the multiplicative effect on the forecast error (PBL) of a particular urbanicity level compared to the Large Metropolitan (LM) areas within each phase. The relative effect is represented by *exp(Coef)* and computed as $e^{\beta_i + \delta_{ij}}$ with coefficients from Eq 4. To evaluate better the relative effect, we also discuss the percentage change in forecast error when compared to LM areas for each phase (*% Diff from LM*). This change is computed as $(1 - e^{\beta_i + \delta_{ij}}) * 100\%$ for a given urbanicity level and phase, and it represents the percentage increase or decrease in the forecast error (PBL) with respect to LM areas. All effects should be interpreted as the relative difference compared to LM areas within each specific phase. The relative coefficient significance is evaluated using `linearHypothesis` in R (*car* package [59])

respectively ($p < 0.001$). Phases 1 and 6 maintained similar disparities to the baseline (11.3% and 12.0% higher errors)

SMM areas show a similar but less pronounced pattern of disparities. The baseline prediction errors for SMM areas are 9.1% higher than LM areas. Like MC areas, SMM areas show varying performance across phases, but with generally smaller magnitudes of disparity. The pattern of better performance in Phases 2 and 5 is repeated, with SMM areas showing 4.4% and 6.0% lower errors than LM areas respectively ($p < 0.001$). The highest disparities for SMM areas occur in Phase 3 (8.9% higher errors) and Phase 6 (5.9% higher errors).

**GLM-2c urbanicity and model type.** Table 11 shows the interaction effect analysis for urbanicity levels and types of forecast models. Overall, we observe that MC and SMM suffer from higher PBL errors than LM areas across all types of forecasting models.

The highest performance disparities across model types are observed between MC and LM counties. Specifically, statistical models are the ones with the most pronounced differences, with MC areas experiencing 9.7% higher errors than LM areas; followed by Compartmental models (reference group) at 6.6%, Ensemble models at 5.3%, baseline models at 4.9% and deep learning models at 4.4%. All these differences remain statistically significant ($p < 0.001$).

As seen before for prediction lookaheads and some phases, SMM areas demonstrate a similar pattern but with smaller magnitudes of disparity. Statistical models show the largest disparity between SMM and LM counties, with PBL errors being 4.8% higher in SMM areas. Compartmental (reference group) and baseline model prediction errors for SMM counties are 2.8% higher than LM areas. The disparities are notably smaller for ensemble and deep learning models. Ensemble models have PBL errors 1.8% higher in SMM counties when compared to LM counties, while deep learning models forecasting cases in SMM counties have PBL errors 1.1% higher than LMs.

These findings suggest that deep learning models may be most effective at minimizing urbanization-related disparities in prediction accuracy. Conversely, statistical models appear to amplify these disparities across both MC and SMM areas. The consistent pattern of higher disparities in MC areas compared to SMM areas, regardless of model type, indicates that predictive challenges in less urbanized areas persist across modeling approaches, though their magnitude can be influenced by model selection.

**GLM-2d: Urbanicity and mobility data.** Table 12 represents the interaction analysis between urbanization levels and the use of mobility data in forecasting models. The relative effects analyses show a similar pattern to the other model-data characteristics we have discussed *i.e.,* when compared against LM counties, MC and SMM counties are associated with significantly higher prediction errors across mobility data use approaches (no mobility data, mobility data or mixed model).

MC areas exhibit the largest disparity, with prediction errors 8.8% higher than LM areas when no mobility data is used to train the COVID-19 case forecasting models. This disparity is modestly reduced when mobility data is used, with MC areas showing 5.8% higher errors compared to LM areas, and 6.5% higher errors for mixed models (ensemble of models trained with and without mobility data). Both reductions in disparity are statistically significant ($p < 0.001$).

On the other hand, SMM areas demonstrate a similar pattern but with smaller magnitudes of disparity. Not using mobility causes prediction errors for SMM areas to be 5.6% higher than LM areas. The incorporation of mobility data appears to be more effective in reducing disparities for SMM areas as well, with models using mobility data showing only 1.7% higher errors compared to LM areas. Mixed models, based on CDC ForecastHub ensembles of both models trained with and without mobility data show an intermediate improvement, with 3.1% higher errors compared to LM areas.

**Table 11. GLM-2c: Urbanicity × model type effects relative to LM areas.**

| Variable | Coefficient Estimates | | | Relative Effect | |
|---|---|---|---|---|---|
| | $e^{Coef.}$ (SE) | 95% CI | | $e^{Coef.}$ | % Diff from LM |
| **Micropolitan (MC)** (Compartmental ref.) | 1.066*** (0.001) | [1.063, 1.069] | | 1.066*** | +6.6% |
| × Baseline Models | 0.984*** (0.004) | [0.977, 0.991] | | 1.049*** | +4.9% |
| × Deep Learning | 0.979*** (0.003) | [0.974, 0.985] | | 1.044*** | +4.4% |
| × Ensemble | 0.988*** (0.002) | [0.984, 0.992] | | 1.053*** | +5.3% |
| × Statistical | 1.029*** (0.002) | [1.024, 1.034] | | 1.097*** | +9.7% |
| **Small/Medium Metro (SMM)** (Compartmental ref.) | 1.028*** (0.001) | [1.025, 1.031] | | 1.028*** | +2.8% |
| × Baseline Models | 1.000 (0.004) | [0.992, 1.009] | | 1.028*** | +2.8% |
| × Deep Learning | 0.983*** (0.003) | [0.976, 0.989] | | 1.011*** | +1.1% |
| × Ensemble | 0.990*** (0.002) | [0.986, 0.995] | | 1.018*** | +1.8% |
| × Statistical | 1.019*** (0.003) | [1.014, 1.025] | | 1.048*** | +4.8% |

**Model Statistics:** Pseudo $R^2$ (CS) = 0.462; Log-Likelihood = 5,983,479; N = 1,526,869
**Notes:** $p < 0.001$, $p < 0.01$, $p < 0.05$. *Dependent Variable:* Square root PBL. *Link Function:* Log. *Regression Family:* Gaussian. The table shows the GLM coefficients and their significance for model GLM-2c. We only discuss urbanicity and model type interaction coefficients. For clarity purposes, all other control variables: Health outcomes, age 65+ and state fixed effects are only shown and discussed in Tables 19, 20 in S1 Appendix. Model Diagnostics are provided in Fig 6c in S1 Appendix. For the Coefficient Estimates, *exp(Coef)* represents the multiplicative effect on the outcome, SE: Standard Error and CI: Confidence Interval. The Relative Effect represents the multiplicative effect on the forecast error (PBL) of a particular urbanicity level compared to the Large Metropolitan (LM) areas within each model type.

The relative effect is represented by *exp(Coef)* and computed as $e^{\beta_i + \delta_{ij}}$ with coefficients from Eq 4. To evaluate better the relative effect, we also discuss the percentage change in forecast error when compared to LM areas for each model type (*% Diff from LM*). This change is computed as $\left(1 - e^{\beta_i + \delta_{ij}}\right) * 100\%$ for a given urbanicity level and model type, and it represents the percentage increase or decrease in the forecast error (PBL) with respect to LM areas. All effects should be interpreted as the relative difference compared to LM areas within each specific model type. The relative coefficient significance is evaluated using `linearHypothesis` in R (*car* package [59])

**Table 12. GLM-2d: Urbanicity × mobility usage effects relative to LM.**

| Variable | Coefficient Estimates | | | Relative Effect | |
|---|---|---|---|---|---|
| | $e^{Coef.}$ (SE) | 95% CI | | $e^{Coef.}$ | % Diff from Rural |
| **Micropolitan (MC)** (No Mobility ref.) | 1.088*** (0.002) | [1.084, 1.092] | | 1.088*** | +8.8% |
| × Mixed | 0.979*** (0.003) | [0.974, 0.984] | | 1.065*** | +6.5% |
| × Mobility Used | 0.972*** (0.002) | [0.968, 0.976] | | 1.058*** | +5.8% |
| **Small/Medium Metro(SMM)** (No Mobility ref.) | 1.056*** (0.002) | [1.052, 1.061] | | 1.056*** | +5.6% |
| × Mixed | 0.976*** (0.003) | [0.970, 0.981] | | 1.031*** | +3.1% |
| × Mobility Used | 0.963*** (0.002) | [0.958, 0.967] | | 1.017*** | +1.7% |

**Model Statistics:** Pseudo $R^2$ (CS) = 0.462; Log-Likelihood = 5,983,425; N = 1,526,869
**Notes:** ***$p < 0.001$, **$p < 0.01$, *$p < 0.05$. *Dependent Variable:* Square root PBL. *Link Function:* Log. *Regression Family:* Gaussian. The table shows the GLM coefficients and their significance for model GLM-2d. We only discuss urbanicity and mobility usage interaction coefficients. For clarity purposes, all other control variables: Health outcomes, age 65+ and state fixed effects are only shown and discussed in Tables 21, 22 in S1 Appendix. Model Diagnostics are provided in Fig 6d in S1 Appendix. For the Coefficient Estimates, *exp(Coef)* represents the multiplicative effect on the outcome, SE: Standard Error and CI: Confidence Interval. The Relative Effect represents the multiplicative effect on the forecast error (PBL) of a particular urbanicity level compared to the Large Metropolitan (LM) areas within each mobility usage category. The relative effect is represented by *exp(Coef)* and computed as $e^{\beta_i + \delta_{ij}}$ with coefficients from Eq 4. To evaluate better the relative effect, we also discuss the percentage change in forecast error when compared to LM areas for each mobility usage category (*% Diff from LM*). This change is computed as $\left(1 - e^{\beta_i + \delta_{ij}}\right) * 100\%$ for a given urbanicity level and mobility usage category, and it represents the percentage increase or decrease in the forecast error (PBL) with respect to LM areas. All effects should be interpreted as the relative difference compared to LM areas within each specific mobility usage category. The relative coefficient significance is evaluated using `linearHypothesis` in R (*car* package [59])

**Summary**: Urbanicity X Model-Data Characteristics

⇒ SMM and MC are consistently associated to higher prediction errors compared to LM across most configurations.
⇒ Disparities decrease with longer forecast horizons.
⇒ Phases 2 and 5 saw a reversal in the pattern with SMM and MC performing better than LM counties.
⇒ Deep learning models demonstrated most balanced performance across urbanization levels.
⇒ Mobility data usage generally helps reduce prediction disparities

## Dashboard

This paper has revealed significant disparities in COVID-19 case prediction accuracy across race, ethnicity and urbanization level. The regression analyses we have presented, evaluate the relationship between COVID-19 case prediction errors and racial and ethnic groups, urbanization level, model type, the use of mobility data, lookahead, and phase. Our findings are based on global trends across all Forecast Hub models, and provide general recommendations for researchers working in COVID-19 prediction models and for decision makers using case predictions to inform pandemic policies. For example, we have shown that mobility data helps reduce the forecast error disparities between racial groups and urbanization levels. This fact can be used modelers and decision makers to support the use of mobility data to enhance forecast models.

Nevertheless, it is important to acknowledge that researchers and decision makers might also want to assess the specific performance of each COVID-19 county case prediction model individually, exploring PBL error differences between racial and ethnic groups or urbanization levels for a given model, their statistical significance, or whether these differences persist when considering specific lookaheads or phases. To enable individual model evaluation, we have created an interactive dashboard (see Fig 3), that is publicly available at https://public.tableau.com/app/profile/saad.mohammad.abrar/viz/fairnessDashboard/FAIRNESSEVALUATIONOFCDCFORECASTHUB. The dashboard displays a model's performance error (PBL) for a given protected attribute - race and ethnicity or urbanization level - that can be selected by the user from the user interface.

To allow for meaningful explorations, the individual model errors are displayed using the Accuracy Equality Ratio (AER) [60], which measures the difference in error distributions between protected and unprotected groups for a given protected attribute. The AER is computed as a quotient between the model's performance error (PBL) for a given protected group $g$ across all counties and the model's performance error for the unprotected group across all counties: $AER_g = \frac{PBL(protected\_group\_g)}{PBL(unprotected\_group)}$ where $PBL(protected\_group\_g)$ and $PBL(unprotected\_group)$ are the pinball ball loss metric for protected and unprotected groups respectively. For the race and ethnicity protected attribute we define three protected groups with respect to the White unprotected group: Asian, Black and Hispanic, and associate the plurality race to each county, i.e., the race or ethnicity that makes up the largest percentage for that county. For the urbanization code, we use the protected groups described in the paper: Micropolitan (MC) and Small and Medium Metro Areas (SMM) with Large Metropolitan Areas being the unprotected group. Similar error distributions between the protected and the unprotected groups will produce AER values close to one. AER values larger than one point to higher errors for the protected group, and AER values smaller than one point to higher errors for White or large metropolitan counties (baseline groups).

**Interactive Features.** The dashboard allows users to carry out different types of racial/ethnic and urbanization fairness analyses for each individual model. As shown in the four boxes at the bottom of Fig 3, the dashboard includes the following key interactive features:

- **Variables of Interest Selection**: Users can choose from six different analytical perspectives to analyze the fairness of a given COVID-19 forecast model:
  – Race or ethnicity analysis, that allows to analyze the fairness of the predictions for a given minority race with respect to White (see example in Fig 3)
  – Urbanicity analysis, that allow to evaluate the fairness of the predictions for a given urbanization level with respect to large metropolitan areas (see Fig 7 in the S1 Appendix)
  – Analysis at the intersection of race/ethnicity and lookahead periods or pandemic phases, that allows to evaluate differences in COVID-19 forecast fairness for a given minority race (with respect to White) and for a given lookahead or pandemic phase (see Figs 8-9 in S1 Appendix for a couple of examples)
  – Urbanicity and lookahead periods or pandemic phases analysis, allowing users to explore differences in COVID-19 forecast fairness for a given urbanization level (with respect to large metropolitan areas) and a given lookahead or phase (see Figs 10-11 in S1 Appendix for a couple of examples)
- **Protected Variable Selection**: The dashboard allows users to focus on specific demographic groups:
  – For racial/ethnic analysis: Black, Hispanic, or Asian AER (White is the baseline group)
  – For urbanicity analysis: Micropolitan or Small/Medium Metro AER (Large Metropolitan is the baseline group)
- **Temporal Analysis Options**:
  – Phase selection (0-6) for analyzing performance across different pandemic periods
  – Lookahead periods (7, 14, 21, or 28 days) for examining how prediction fairness varies with forecast horizon

**Sample Use Case.** Fig 3 shows an example of the dashboard for the exploration of individual model performance by race and type of model, with a focus on the relationship between the prediction errors for Hispanic and White counties ($AER_{Hispanic}$). The box plots for each model represent its AER distribution across all counties; and a user can explore the median AER as well as its quantiles for each predictive model. In this example, most of the AERs are above one, pointing to unfair forecasts (higher errors) for the Hispanic group when compared to White counties.

Hovering over the model points displays all the information in the format of a 'fairness nutritional card' as shown in Fig 4. 'Nutritional labels' were proposed by Stoyanovich and Howe to assess model fairness [50] drawing an analogy to the food industry, where simple, standard labels convey information about the ingredients and production processes. We have adapted these cards to the COVID-19 fairness context, as a way to provide detailed COVID-19 forecast model fairness information. The fairness nutritional cards in our dashboard (see Fig 4 for a sample) provide detailed information organized into four key sections: (1) *Model Information*, which identifies the team name and the variables being analyzed; (2) *Mean Difference with Unprotected Reference Group*, which quantifies the prediction error differences between protected and unprotected groups in terms of PBL values, including upper and lower bounds; (3) *Team AER Values*, showing both the median and specific AER values that indicate the relative performance between groups; and (4) Coverage Info, which provides context about the number of counties and total predictions covered by each team.

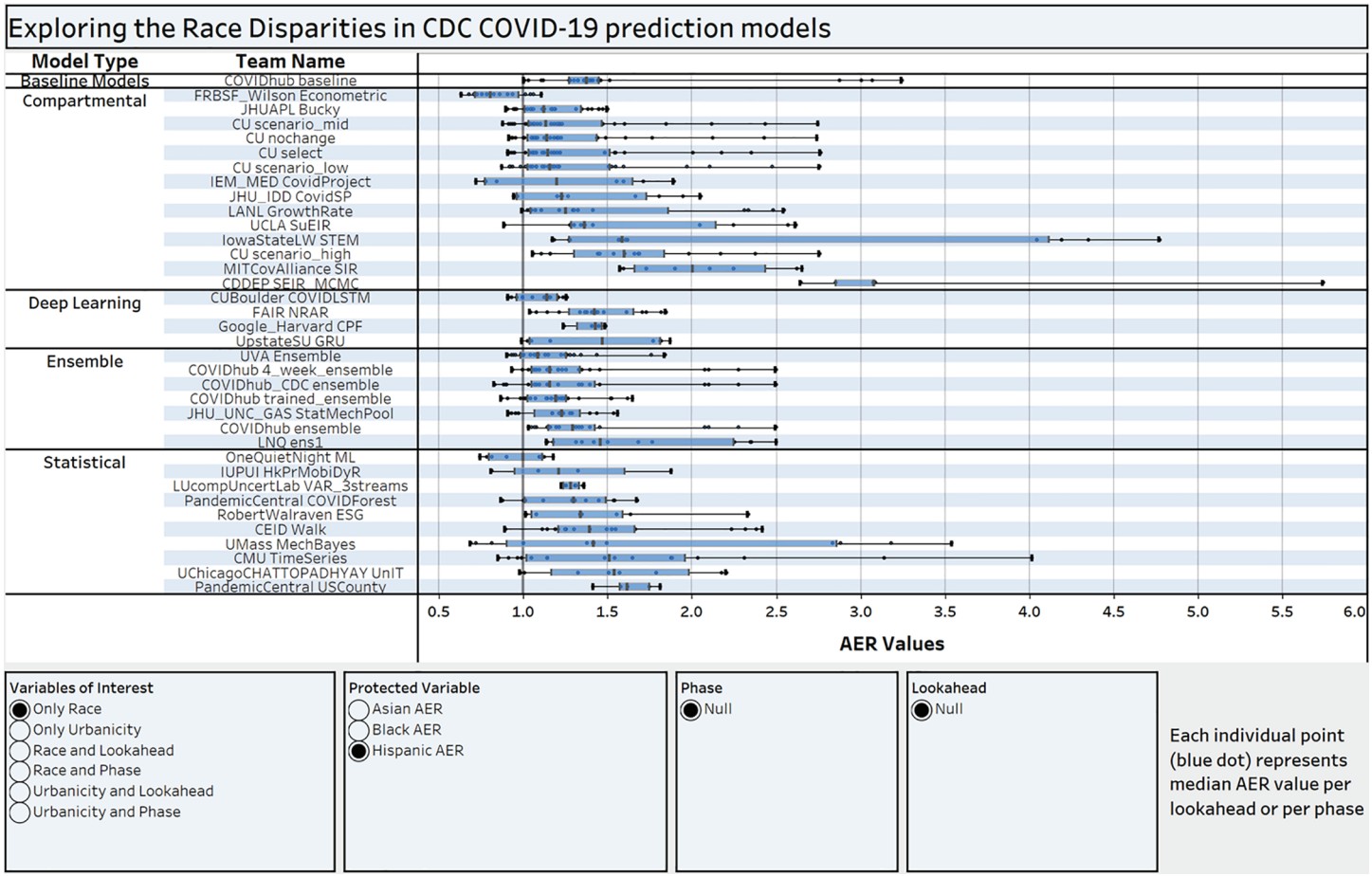

**Fig 3. Forecast hub fairness dashboard showing the average error ratio (AER) distribution across different COVID-19 prediction models, organized by model type.** Within each model type, teams are sorted in ascending order based on their median AER values. Since the user has selected "Only Race" as the variable of interest (see bottom left box) and "Hispanic" as the protected variable (see bottom center box), the AER values compare prediction errors between Hispanic and White counties, where values above 1.0 indicate higher prediction errors for Hispanic counties. Box plots show the distribution of AER values across all predictions, with the center line representing the median, boxes showing the interquartile range, and whiskers extending to the minimum and maximum values.

Looking into Fig 3 and Fig 4, we can observe that the *IowaStateLW STEM* model exhibits a wide range of median AER values (min: 1.172, max: 4.772, median: 1.588) across its predictions, indicating highly variable fairness performance when assessed across different phases and lookaheads. On the other hand, the *LUcompUncertLab VAR_3streams* model shows consistent median AER values within a small range from 1.228 to 1.334 (median: 1.280). Comparing both models, we observe that although both of them are systematically producing predictions that are less fair for Hispanic counties (AER values are larger than 1), the *LUcompUncertLab VAR_3streams* model has lower max AER values, suggesting that model might be a better choice.

Overall, we posit that the dashboard facilitates dynamic exploration of a diverse set of metrics across different dimensions, allowing users to examine how fairness measures vary with changes in pandemic phases and lookahead through the user interface. When exploring these temporal variables, the nutritional card automatically updates to display the relevant phase or lookahead information for the selected view.

**A.**

MODEL FAIRNESS CARD

Model Information
Team Name:                                          LUcompUncertLab VAR_3streams
Variables of Interest:                              Only Race
--------------------------------------------------------------------------------

Mean Difference with Unprotected Reference Group
Mean Differences in PBL with unprotected group:  0.0000748
Upper Difference:                                   0.000274
Lower Difference:                                   -0.0001243
--------------------------------------------------------------------------------

Team AER Values
Median AER Value:                                   1.280
AER Value:                                          1.239
--------------------------------------------------------------------------------

Coverage Info
Total counties coverage                             3143
Total Number of Predictions                         53284

**B.**

MODEL FAIRNESS CARD

Model Information
Team Name:                                          IowaStateLW STEM
Variables of Interest:                              Only Race
--------------------------------------------------------------------------------

Mean Difference with Unprotected Reference Group
Mean Differences in PBL with unprotected group:  0.0002463
Upper Difference:                                   0.000446
Lower Difference:                                   0.0000463
--------------------------------------------------------------------------------

Team AER Values
Median AER Value:                                   1.588
AER Value:                                          1.615
--------------------------------------------------------------------------------

Coverage Info
Total counties coverage                             3104
Total Number of Predictions                         80704

**Fig 4. Model fairness card displaying key performance metrics including model information, prediction error differences between protected and unprotected groups, AER values, and coverage statistics.**

## Discussion

This study highlights the critical need to audit COVID-19 prediction models due to significant disparities in prediction accuracy. Our findings reveal that certain minority groups, especially Hispanic communities, and less urbanized areas consistently experience higher prediction errors. The race and ethnicity analysis revealed that increases in Hispanic population (when compared to White) exhibit significantly higher PBL errors; while increases in Asian and Black population are associated with lower PBL errors when compared to the White population (reference group), and while controlling for health outcomes and older population at the county level. The analysis on urbanization levels, on the other hand, revealed an inverse relationship between the level of urbanization and the magnitude of prediction errors underscoring the unique challenges encountered by rural areas. Rural counties consistently face higher prediction errors than their urban counterparts, a pattern that persists across various model types and forecast windows.

*Potential Reasons.* These findings could be related to data quality (COVID-19 cases, mobility data) or structural problems. Hispanic groups or less urbanized areas being associated with significantly higher errors (when compared to their baselines: White population and large metropolitan area), could point to lack of quality COVID-19 case or mobility data for the Hispanic population and less urbanized areas. It could also be due to more complex spreading patterns that make COVID-19 cases harder to predict for the Hispanic population or rural areas, or two structural differences such as reduced access to medical facilities or testing sites. On the other hand, these findings are also pointing to better COVID-19 case or mobility data for Asian and Black population when compared to White; or to simpler spreading patterns that are easier to forecast, hence producing lower errors.

The implications of these findings are significant, since systematic disparities in model performance could lead to unfair distribution of public health resources or to less effective pandemic response efforts in Hispanic counties and in less densely populated regions, when compared to White and urban regions. Our definition of prediction fairness is focused on achieving similar prediction errors across racial, ethnic and urban-rural groups because COVID-19 cases have been used to make resource allocation and intervention decisions e.g.,

hospital beds or stay-at-home orders. Hence, higher prediction errors for minority racial groups or rural regions could in turn translate into unfair resource allocation for communities that have borne the brunt of the pandemic. Ultimately, we want to ensure our findings serve as a critical call to action for researchers and decision makers to analyze model performance disaggregated by racial/ethnic and urban-rural variables.

**Interaction Analysis**. Our interaction analysis provides a more multifaceted understanding of fairness in COVID-19 modeling. We find that deep learning models tend to produce the lowest disparities in errors across racial, ethnic and urban-rural groups, while compartmental and statistical models tend to be associated with the highest disparities. Our results have also shown that the use of mobility data helps reduce prediction error disparities for racial and ethnic groups as well as across urbanization levels. Short-term lookaheads and certain pandemic phases (case-peak phases 1, 3 and 4) are also associated with higher prediction error disparities for minority racial groups and rural areas. These findings highlight the complex interplay between model characteristics, data inputs, and social determinants in shaping prediction fairness.

*Potential Reasons.* Decreasing disparities in prediction errors for higher lookaheads could be due to a reduction in the effect of COVID-19 case data bias (positive or negative) for minority groups and less urbanized areas. In fact, as the lookahead increases, prior work has shown that case prediction becomes more difficult [33], and this forecasting complexity appears to have a stronger effect than data bias on the prediction errors, thus making all errors more similar (more fair) across racial/ethnic groups and urbanization levels. When looking into prediction error disparities across phases and race/ethnicity or urbanization levels, we argue that the higher prediction differences tend to take place during initial phases of the pandemic, which could point to factors like initial data collection issues, testing accessibility, or reporting practices that may have varied across racial and ethnic groups during the early stages of the pandemic, especially for Hispanic groups and less urbanized areas. It also revealing of the fact that there were very limited historical COVID-19 case data to learn from in the early phases.

Looking into types of models, deep learning (DL) models appear to be the best choice to reduce performance error differences between LMs and less urbanized areas; while DL models also appear to be a good compromise across minority groups. This could be pointing to DL modeling being able to better capture spatio-temporal dependencies without the behavioral assumptions of compartmental or statistical models, that show higher performance differences for some racial groups and across less urbanized levels.

Finally, mobility data appears to be providing additional information (insights into behavioral patterns) that helps reduce biases in model performance across the aforementioned sensitive groups.

These interaction results highlight the need for researchers and modelers to carefully examine their data sources, model assumptions, and potential biases that could lead to unfair predictions for certain population groups. Incorporating fairness considerations into the model development, validation, and deployment processes is essential to ensure equitable outcomes. Public health officials and policymakers should be aware of the potential disparities in the accuracy of COVID-19 prediction models and work closely with modelers to mitigate these disparities. Failure to address these issues could lead to the perpetuation of health inequities and could eventually undermine the effectiveness of pandemic response efforts.

In addition to our findings, this study has several limitations that should be acknowledged. Firstly, we had to exclude some U.S. counties from our analysis due to insufficient data availability for the data sources we used. Second, while our work primarily focused on urbanicity

and race/ethnicity as fairness-related variables, other important attributes, such as socioeconomic status or access to healthcare, etc, were not considered and could be explored in future research. Additionally, we were unable to incorporate all minority racial groups like AIAN and NHPI due to inadequate population sizes, which constrained our ability to assess fairness comprehensively across all demographic groups.

**Moving Forward.** In our study, we have focused exclusively on county-level predictions because these are closer to local realities and allow for more actionable decision-making than state-level predictions. However, county-level statistics were collected only for COVID-19 cases, with hospitalizations or deaths only accounted for at the state level. Since prior work has shown that case counts might be more biased than hospitalization or death statistics [3], the results reflected in this paper could potentially change if hospitalization or death data were available at the county level and this study was replicated.

We posit that future research in COVID-19 case prediction models should focus on developing and validating bias mitigation strategies that account for performance disparities across race, ethnicity and urbanization levels. This may involve exploring alternative data sources, refining model architectures, and incorporating techniques to ensure fairness across different population groups. Additionally, more comprehensive and standardized race and ethnicity data collection in public health surveillance systems is crucial to enable accurate assessments of model fairness and to guide equitable decision-making.

## Conclusions

Our paper shows significant diverse predictive performance across social determinants for the Forest Hub COVID-19 models, with some minority racial and ethnic counties as well as less urbanized counties often associated with statistically significant higher prediction errors. We also show that these higher errors are often times present for specific model types, lookaheads and pandemic phases; and that these findings generally hold across different race associations. **We hope this paper will encourage Forecast Hub modelers, the CDC and COVID-19 modelers to report fairness metrics together with accuracy, and to reflect on the potential negative impacts of the models on specific social groups and contexts.**

## Acknowledgments

I would like to thank Dr. Peter Rankel for the statistical inputs for the paper.

We gratefully acknowledge the contributions of all research teams that have shared their models through the COVID-19 Forecast Hub, including Columbia University Projections (CU-Nochange, CU-scenario high, CU-scenario mid, CU-scenario low, CU-select) [8]; CMU-TimeSeries [61]; COVIDHub Models (COVIDhub CDC-ensemble, COVIDhub-trained ensemble, COVIDhub-ensemble, COVIDhub-4 week ensemble, COVIDhub-baseline) [62,63]; IEM MED-CovidProject [64]; LNQ-ens1 [65]; JHU IDD-CovidSP [66]; CUBoulder-COVIDLSTM [6]; Google Harvard-CPF [5]; FRBSF Wilson-Econometric [67]; UMass-MechBayes [68]; LANL-GrowthRate [69]; UCLA-SuEIR [9]; CEID-Walk [70]; RobertWalraven-ESG [71]; OneQuietNight-ML [72]; IUPUI-HkPrMobiDyR [10]; UpstateSU-GRU [4]; UChicagoCHATTOPADHYAY-UnIT [73]; PandemicCentral (PandemicCentral-USCounty, PandemicCentral-COVIDForest) [11]; JHUAPL-Bucky [74]; LUcompUncertLab-VAR 3streams [75]; FAIR-NRAR [7]; JHU UNC GAS-StatMechPool [76]; IowaStateLW-STEM [77]; MITCovAlliance-SIR [78]; UVA-Ensemble [12];

## Supporting information

**S1 Appendix. This appendix provides supplementary information to facilitate a deeper understanding and interpretation of the results.**
(PDF)

## Author contributions

**Conceptualization:** Saad Mohammad Abrar, Vanessa Frias Martinez.

**Data curation:** Saad Mohammad Abrar.

**Formal analysis:** Saad Mohammad Abrar.

**Funding acquisition:** Vanessa Frias Martinez.

**Investigation:** Nekabari Sigalo.

**Methodology:** Saad Mohammad Abrar, Nekabari Sigalo, Vanessa Frias Martinez.

**Project administration:** Vanessa Frias Martinez.

**Resources:** Saad Mohammad Abrar, Naman Awasthi, Daniel Smolyak.

**Software:** Saad Mohammad Abrar.

**Supervision:** Vanessa Frias Martinez.

**Validation:** Saad Mohammad Abrar, Naman Awasthi, Daniel Smolyak, Vanessa Frias Martinez.

**Visualization:** Saad Mohammad Abrar, Vanessa Frias Martinez.

**Writing – original draft:** Saad Mohammad Abrar, Vanessa Frias Martinez.

**Writing – review & editing:** Saad Mohammad Abrar, Naman Awasthi, Daniel Smolyak, Vanessa Frias Martinez.

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
