## [Decision Letter · Decision Letter 0]

24 Sep 2024

PONE-D-24-32720Auditing the Fairness of COVID-19 Forecast Hub's Case Prediction ModelsPLOS ONE

Dear Dr. Abrar,

Thank you for submitting your manuscript to PLOS ONE. After careful consideration, we feel that it has merit but does not fully meet PLOS ONE’s publication criteria as it currently stands. Therefore, we invite you to submit a revised version of the manuscript that addresses the points raised during the review process.

Specifically, the following two of PLOS ONE's publication criteria need to be addressed in your revised manuscript:"**3. Experiments, statistics, and other analyses are performed to a hihg technical standard and are described in sufficient detail**.""**5. The article is presented in an inteligible fashion and is written in standard English.**" This manuscript provides informative insights and has the potential to benefit the readers of PLOS ONE. However, there is no statement regaridng the availability of data and code used in this study, which is a critical requirement for modeling studies. Replicability is crucial in predictive modeling research, as it guarantees the integrity of the study. So, please ensure this information is included. Additionally, please revise the tone of the manuscript to make it more neutral and scientific. While PLOS ONE does not have a word count limit for manuscripts, the current version is somewhat lengthy. Please reduce repetative language and improve the overall readability of this manuscript. I encourage you to revise it to make it both neat and informative.

We look forward to receiving your revised manuscript.

Kind regards,

Weijun Yu, Ph.D., M.D., M.S.

Academic Editor

PLOS ONE

“This work has been funded with a National Science Foundation grants, NSF #1750102, NSF

#2210572”

4. We note that Figures 2 and 3 in "SI_1.pdf" in your submission contain [map/satellite] images which may be copyrighted. All PLOS content is published under the Creative Commons Attribution License (CC BY 4.0), which means that the manuscript, images, and Supporting Information files will be freely available online, and any third party is permitted to access, download, copy, distribute, and use these materials in any way, even commercially, with proper attribution. For these reasons, we cannot publish previously copyrighted maps or satellite images created using proprietary data, such as Google software (Google Maps, Street View, and Earth). For more information, see our copyright guidelines: http://journals.plos.org/plosone/s/licenses-and-copyright.

a. You may seek permission from the original copyright holder of Figures 2 and 3 in "SI_1.pdf" to publish the content specifically under the CC BY 4.0 license. 

Reviewers' comments:

Reviewer's Responses to Questions

**Comments to the Author**

1. Is the manuscript technically sound, and do the data support the conclusions?

Reviewer #1: Partly

Reviewer #2: Yes

Reviewer #3: Yes

2. Has the statistical analysis been performed appropriately and rigorously? 

Reviewer #1: Yes

Reviewer #2: Yes

Reviewer #3: No

3. Have the authors made all data underlying the findings in their manuscript fully available?

Reviewer #1: No

Reviewer #2: No

Reviewer #3: Yes

4. Is the manuscript presented in an intelligible fashion and written in standard English?

Reviewer #1: Yes

Reviewer #2: Yes

Reviewer #3: Yes

5. Review Comments to the Author

Reviewer #1: The statistics seem sound and the justification of the study is clear but I have questions and comments about the fairness card and dashboard.

The manuscript should include a data availability statement. I think it is reasonable not to provide access to the dashboard before publication but it more explanation is required for the review.

Fairness nutritional card – in what way is this study about food?

Inconsistency of “COVID-19” and “covid-19”.

The dashboard should be self-explanatory and easy to read.

(i) The meanings of the dots, blue shading, horizontal lines between dots (and no horizontal lines between dots) for the AERs should be explained

(ii) It is not clear what the plot would look like with Race and Lookahead or Race with phase (reporting more than one view of the dashboard would be useful).

(iii) Adding the dates of data would be useful to give context to the data.

(iv) It would be helpful to give a simple example of how the dashboard would be used (and therefore demonstrate its usefulness) e.g. interpreting IowaStateLW STEM (wide range of AERs) vs LUcompUncertLab Var_3streams (narrow and lower range of AERs) for “Only Race” and “Hispanic AER” variable i.e. the latter model scores higher on fairness?

(v) Why are the Phase and Lookahead options necessary as options for Variables of Interest seem to cover both ?

(vi) Spaces between the model names would be helpful as it is currently difficult to match the AERs to the models

(vii) It would be helpful to provide options to: (a) sort the models in a different way e.g by fairness based on the upper limit of the AER trace; (b) subset models e.g. all with *** (shading out the others); and (c) highlight individual models

Reviewer #2: The authors present a critical study of several COVID-19 prediction models that utilize U.S. data from the Forecast Hub COVID-19 platform. They propose incorporating race, ethnicity, and rural-urban characteristics to assess the predictions of 36 models, categorized as ensemble, statistical, compartmental, deep learning, and baseline models. Their analysis shows that the models do not consistently perform well across different social determinants.

The study provides results based on various scenarios, including mobility data, lookaheads, and different COVID-19 phases. The analytical approach is clearly explained and appears reasonable.

I suggest addressing the following issue before publication:

First of all, I suggest to the authors soften some of the statements that come across as harsh regarding the performance of the various models. Many researchers developed these models quickly in order to provide insights into the dynamics of the rapidly spreading pandemic. Most of these researchers contributed to the development of predictive models, even while working on other research projects, due to the global urgency. Therefore, I recommend that the authors avoid a polemical tone in the article, considering the historical context in which the models were proposed. Instead, they should encourage more precise research in the future without undermining the efforts made by researchers during the spread of COVID-19. In any case, I believe that the efforts of numerous researchers around the world during this tragic period should be appreciated.

In addition:

1 Clarify the Focus on U.S. Data: The paper should explicitly state that the comparisons are based on U.S. data and models. There is also a European Forecast Hub producing similar predictions. However, in Europe, particularly in countries like Italy, race differences may be less pronounced due to the public healthcare system. It is not clearly specified in the paper that the analysis pertains to U.S. data and models.

2 Reduce Repetitions: Some sentences are repeated multiple times throughout the paper. After reading the entire paper, I suggest revising these repetitions for clarity and conciseness.

3 Ensure Replicability and possibly reproducibility: To make this research replicable, please confirm whether the data and code used for the analysis are publicly available.

4 It would be more appropriate to cite each model presented in Figure 8 and in the Appendix with proper references to the authors. These citations should be available on the Forecast Hub website.

Reviewer #3: The manuscript provides a valuable case study on applying fairness to evaluate COVID-19 models in the United States. This is an important contribution, addressing a significant gap in the current modeling literature. The work is commendable in its attempt to assess the performance of models. However, the paper is notably long for an epidemiological study, and the authors might consider condensing certain sections to enhance their message.

Additionally, I offer the following observations for consideration:

1) In the abstract, the phrase “we show […] performance across social determinants, with minority racial and ethnic groups as well as less urbanized areas” suggests the inclusion of multiple social determinants. However, the manuscript primarily focuses on racial factors. Please clarify if other social determinants were indeed analyzed or revise the wording.

2) In line 54, the authors assert that self-reported race is the “most accurate source and prevents errors.” While I agree, this warrants a citation.

3) The classification of model types requires further elaboration. The categories used are not immediately intuitive. For instance (as I understand it) the IEM_MED-CovidProject, incorporates deep learning into a compartmental model.

4) The criteria used to delineate COVID-19 phases should be expanded upon. For example, Phases 5 and 6 appear to represent the same pandemic wave. A clearer justification for these phase distinctions is necessary.

5) In the Plurality Approach, why were the Non-White used in the Majority dropped? Please elaborate. There seems to be an invisibilization of some minorities (such as Native Americans) that under this approach are converted to white (look at Alaska, for example). A paper on fairness should not exclude minorities without a careful explanation.

6) The rationale for selecting specific variables in the PBL model is unclear. COPD and age65+ are reasonable covariates in the context of COVID-19, but other relevant factors, such as additional chronic conditions (e.g., obesity) or sex, are omitted. The authors should provide a more detailed epidemiological rationale for the inclusion of these covariates and the exclusion of others.

7) Similarly, it is unclear why race is the only covariate considered in the fairness analysis. The manuscript might be more accurately described as addressing racial fairness rather than fairness (broadly). Fairness-related variables, such as gender/sex, poverty, social vulnerability, and age, are not considered.

8) It would be useful to include more detailed information in the appendix to assess model fit, such as ANCOVA tables with corresponding statistics. The finding that all regression coefficients were statistically significant is intriguing…

9) In a study centered on fairness, it is worth highlighting the exclusion of U.S. territories like Puerto Rico from the evaluation. The manuscript appears to exclude Puerto Rico from both the U.S. map and the county count. This exclusion omits a significant Hispanic population, and the authors should explain the rationale for this decision.

10) This is related to 8) in that ANCOVA's assumptions aren't tested for (e.g. linearlty, normality etc) in each of the ANCOVAs. ANCOVA is a model and it has its assumptions. If the assumptions aren't met we, readers, cannot conclude whether what we are seeing as results is correct or not. You can consult "ANOVA and ANCOVA: a GLM" approach for example.

11) Lastly, there is an inconsistency in the number of counties reported in the Supplementary Appendix. Table 2 lists 3,144 counties, while Tables 3 and 4 list 3,142 counties. Please clarify the discrepancy.

6. PLOS authors have the option to publish the peer review history of their article (what does this mean?). If published, this will include your full peer review and any attached files.

Reviewer #1: No

Reviewer #2: No

Reviewer #3: **Yes: **Rodrigo Zepeda-Tello

---

## [Author Response · Author response to Decision Letter 1]

8 Jan 2025

We have attached a separate file addressing the questions as well.

Reviewer #1:

We thank the reviewer for the critical feedback provided. We have responded to each individual comment in this document, and we have updated the main manuscript and the appendix accordingly.

The statistics seem sound and the justification of the study is clear but I have questions and comments about the fairness card and dashboard.

The manuscript should include a data availability statement. I think it is reasonable not to provide access to the dashboard before publication but it more explanation is required for the review.

The datasets used in this article are publicly available in the following repositories:

1. All COVID-19 Forecasts are available at: https://github.com/reichlab/covid19-forecast-hub/tree/master

2. All COVID-19 Ground Truth Data are available at: https://github.com/CSSEGISandData/COVID-19

3. All Race and Ethnicity, Population Data are available at: https://www.census.gov/data/tables/time-series/demo/popest/2020s-counties-detail.html

4. The Urban-Rural Classification Scheme for Counties: https://www.cdc.gov/nchs/data\_access/urban\_rural.htm\#2013\_Urban-Rural\_Classification\_Scheme\_for\_Counties

5. All Health Outcome Datasets are provided at: https://data.cdc.gov/500-Cities-Places/PLACES-Local-Data-for-Better-Health-Place-Data-202/eav7-hnsx/about_data

All datasets used are publicly available. We have added the data availability statement in the submission portal, and all links are also available in the paper as references.

Fairness nutritional card – in what way is this study about food?

We have clarified this in the new version of the paper (section Dashboard). 'Nutritional labels' were proposed by Stoyanovich and Howe to assess model fairness [1] drawing an analogy to the food industry, where simple, standard labels convey information about the ingredients and production processes. We have adapted these cards to the COVID-19 fairness context, as a way to provide detailed COVID-19 forecast model fairness information.

[1] Stoyanovich J, Howe B. Nutritional labels for data and models. A Quarterly bulletin of the Computer Society of the IEEE Technical Committee on Data Engineering. 2019;42(3)

Inconsistency of “COVID-19” and “covid-19”.

We have addressed and removed the inconsistencies.

The dashboard should be self-explanatory and easy to read.

(i) The meanings of the dots, blue shading, horizontal lines between dots (and no horizontal lines between dots) for the AERs should be explained

We have extensively edited the Dashboard section to clarify its meaning and use. In addition, we have clearly explained the meaning of the different elements raised by the reviewer in the visualization in Figure 3 (legend). Specifically:

For each individual model, we compute the median AER for each lookahead and each phase. Each individual dot represents the median AER value per lookahead or per phase.

The blue shading shows the interquartile range (IQR), containing 50% of the model's AER values

The central vertical line within the blue shading indicates the overall median AER across all phases and lookaheads.

Thin vertical lines (whiskers) extend to show the minimum and maximum AER values, excluding statistical outliers.

(ii) It is not clear what the plot would look like with Race and Lookahead or Race with phase (reporting more than one view of the dashboard would be useful).

We have added a description of the different types of Dashboard views available in the main paper (“Dashboard” section). In addition, we have added plots for each visualization type in the S1 Appendix (with links to them in the main paper, see Figures 7-11).

(iii) Adding the dates of data would be useful to give context to the data.

Predictions in the dashboard can be explored by phase, with phases being defined by COVID-19 case volume patterns as described in section “Model-Data Characteristics” in page 6. The phases themselves capture the meaningful temporal patterns in the pandemic's progression - from periods of low case volumes to peak waves - which is more relevant for understanding how model fairness varied across different epidemiological contexts than calendar dates.

(iv) It would be helpful to give a simple example of how the dashboard would be used (and therefore demonstrate its usefulness) e.g. interpreting IowaStateLW STEM (wide range of AERs) vs LUcompUncertLab Var_3streams (narrow and lower range of AERs) for “Only Race” and “Hispanic AER” variable i.e. the latter model scores higher on fairness?

To illustrate the dashboard's utility in assessing model fairness, we have extensively edited the Dashboard section with a subsection called “Sample Use Case” (page 24). In this sample use case, we explore how the dashboard and the nutritional cards (Figures 3 and 4) can be used to make decisions about the fairness of the models.

For the specific example raised by the reviewer, we have added the following text to the main paper: “The IowaStateLW STEM model exhibits a wide range of median AER values (min: 1.172, max: 4.772, median: 1.588) across its predictions, indicating highly variable fairness performance when assessed across different phases and lookaheads. On the other hand, the LUcompUncertLab VAR_3streams model shows consistent median AER values within a small range from 1.228 to 1.334 (median: 1.280). Comparing both models, we observe that although both of them are systematically producing predictions that are less fair for Hispanic counties (AER values are larger than 1), the LUcompUncertLab VAR_3streams model has lower max AER values, suggesting that model might be a better choice.”

It is important to note that while the box plots reveal the distribution of median AER values (calculated separately for each phase or lookahead period), direct comparisons between models should be made cautiously as they may have predictions for different numbers of phases (for example IowaStateLW STEM has predictions for Phase 0,1,2 and LUcompUncertLab VAR_3streams has predictions for Phase 5,6). Hence, we recommend doing this analysis in combination with Figure 1 in S1 Appendix. Overall, this type of analysis helps researchers and decision-makers understand individual model's fairness implications across different demographic groups, providing insights beyond traditional accuracy metrics.

(v) Why are the Phase and Lookahead options necessary as options for Variables of Interest seem to cover both ?

The Phase and Lookahead options serve a different purpose from the Variables of Interest selection. While Variables of Interest allows users to examine how AERs vary across all phases or lookaheads (showing the complete distribution), the individual Phase and Lookahead selectors enable users to filter the data to a specific phase or prediction horizon. This granular filtering capability allows researchers to investigate fairness metrics under particular temporal conditions - for example, examining model performance specifically during peak case periods (a particular phase) or for short-term predictions (a specific lookahead). For detailed examples of how these filtering options reveal different patterns in model fairness, we refer the reviewer to the newly edited Dashboard section (page 23) as well as to Figures 8-11 in S1 Appendix, which demonstrate how AER distributions can vary substantially when examined at specific temporal granularities.

(vi) Spaces between the model names would be helpful as it is currently difficult to match the AERs to the models

We have added row banding for clearer distinction between models.

(vii) It would be helpful to provide options to: (a) sort the models in a different way e.g by fairness based on the upper limit of the AER trace; (b) subset models e.g. all with *** (shading out the others); and (c) highlight individual models

We have sorted the models per model type in increasing order of median AER value.

Reviewer #2:

We thank the reviewer for the critical feedback provided. We have responded to each individual comment in this document, and we have updated the main manuscript and the appendix accordingly.

The authors present a critical study of several COVID-19 prediction models that utilize U.S. data from the Forecast Hub COVID-19 platform. They propose incorporating race, ethnicity, and rural-urban characteristics to assess the predictions of 36 models, categorized as ensemble, statistical, compartmental, deep learning, and baseline models. Their analysis shows that the models do not consistently perform well across different social determinants.

The study provides results based on various scenarios, including mobility data, lookaheads, and different COVID-19 phases. The analytical approach is clearly explained and appears reasonable.

I suggest addressing the following issue before publication:

First of all, I suggest to the authors soften some of the statements that come across as harsh regarding the performance of the various models. Many researchers developed these models quickly in order to provide insights into the dynamics of the rapidly spreading pandemic. Most of these researchers contributed to the development of predictive models, even while working on other research projects, due to the global urgency. Therefore, I recommend that the authors avoid a polemical tone in the article, considering the historical context in which the models were proposed. Instead, they should encourage more precise research in the future without undermining the efforts made by researchers during the spread of COVID-19. In any case, I believe that the efforts of numerous researchers around the world during this tragic period should be appreciated.

We agree with the reviewer, and we highly value the work that our colleagues have done in the development of complex COVID-19 forecast models. Hence, we have reframed some of our statements particularly in the introduction, presenting fairness evaluation as an opportunity for enhancement rather than a limitation. For example, we revised: "the Forecast Hub fails to evaluate..." to "While the Forecast Hub has made significant contributions through its accuracy-focused predictions..., there is an opportunity to expand its evaluation framework… Throughout the manuscript, we have maintained our focus on constructively identifying opportunities for improvement while acknowledging the valuable work done under extremely challenging circumstances. We kindly ask the reviewer to check highlighted changes in the Introduction,

In addition:

1 Clarify the Focus on U.S. Data: The paper should explicitly state that the comparisons are based on U.S. data and models. There is also a European Forecast Hub producing similar predictions. However, in Europe, particularly in countries like Italy, race differences may be less pronounced due to the public healthcare system. It is not clearly specified in the paper that the analysis pertains to U.S. data and models.

We have made necessary changes to the title, abstract and main manuscript to specify we have focused our work on the US CDC ForecastHub.

2 Reduce Repetitions: Some sentences are repeated multiple times throughout the paper. After reading the entire paper, I suggest revising these repetitions for clarity and conciseness.

We have extensively edited the paper to reduce repetitions. We have also simplified our analytical approach with a focus on the regression analysis instead of the majority, plurality and regression methods, which added a lot of repetitions.

3 Ensure Replicability and possibly reproducibility: To make this research replicable, please confirm whether the data and code used for the analysis are publicly available.

All the datasets used in this study are publicly available, and we have specified that in our manuscript. The code will be made publicly available upon publication.

The datasets supporting the conclusions of this article are available in the following repositories:

1. All COVID-19 Forecasts are available at: https://github.com/reichlab/covid19-forecast-hub/tree/master

2. All COVID-19 Ground Truth Data are available at: https://github.com/CSSEGISandData/COVID-19

3. All Race and Ethnicity, Population Data are available at: https://www.census.gov/data/tables/time-series/demo/popest/2020s-counties-detail.html

4. The Urban-Rural Classification Scheme for Counties: https://www.cdc.gov/nchs/data\_access/urban\_rural.htm\#2013\_Urban-Rural\_Classification\_Scheme\_for\_Counties

5. All Health Outcome Datasets are provided at: https://data.cdc.gov/500-Cities-Places/PLACES-Local-Data-for-Better-Health-Place-Data-202/eav7-hnsx/about_data

We have also added the data availability statement in the submission portal.

4 It would be more appropriate to cite each model presented in Figure 8 and in the Appendix with proper references to the authors. These citations should be available on the Forecast Hub website.

We have added links to each paper and model in the Acknowledgements section.

Reviewer #3:

We thank the reviewer for the critical feedback provided. We have responded to each individual comment in this document, and we have updated the main manuscript and the appendix accordingly.

We would like to highlight a major change in the analytical approach. As identified by the reviewers, the paper was a bit long and repetitive at times. This was mostly due to the three race association approaches we proposed: majority, plurality and regression. To simplify the paper, and to be able to provide an in-depth analysis of our modeling choices, we have focused only on the regression approach. We kindly ask the reviewer to check our narrowed down approach in the “Analytical Approach” section (pages 6-10).

The manuscript provides a valuable case study on applying fairness to evaluate COVID-19 models in the United States. This is an important contribution, addressing a significant gap in the current modeling literature. The work is commendable in its attempt to assess the performance of models. However, the paper is notably long for an epidemiological study, and

the authors might consider condensing certain sections to enhance their message.

Additionally, I offer the following observations for consideration:

1) In the abstract, the phrase “we show […] performance across social determinants, with minority racial and ethnic groups as well as less urbanized areas” suggests the inclusion of multiple social determinants. However, the manuscript primarily focuses on racial factors. Please clarify if other social determinants were indeed analyzed or revise the wording.

We have updated the paper to clarify that our focus is on two social determinants only: race/ethnicity and urbanization levels.

2) In line 54, the authors assert that self-reported race is the “most accurate source and prevents errors.” While I agree, this warrants a citation.

We have added a relevant citation in line 54: https://pmc.ncbi.nlm.nih.gov/articles/PMC1447261/

3) The classification of model types requires further elaboration. The categories used are not immediately intuitive. For instance (as I understand it) the IEM_MED-CovidProject, incorporates deep learning into a compartmental model.

We have added an extensive discussion of our classification model approach in the S1 Appendix (Section 1.2). We kindly ask the reviewer to check the added explanations.

Regarding the IEM_MED-CovidProject model specifically, while it uses AI techniques for parameter fitting, we classify it as compartmental because its core predictive mechanism relies on a SEIR framework, with AI serving an auxiliary role in parameter optimization. This classification prioritizes the fundamental epidemiological structure over supplementary computational techniques used for implementation. This taxonomy acknowledges that modern epidemiological models often incorporate multiple computational techniques, but classifies them based on their primary predictive mechanism rather than auxiliary methods used for implementation or optimization. We recognize that some models may exhibit hybrid characteristics, but we assign them to categories based on their dominant m

---

## [Decision Letter · Decision Letter 1]

2 Feb 2025

Auditing the Fairness of US COVID-19 Forecast Hub's Case Prediction Models

PONE-D-24-32720R1

Dear Dr. Abrar,

We’re pleased to inform you that your manuscript has been judged scientifically suitable for publication and will be formally accepted for publication once it meets all outstanding technical requirements.

Kind regards,

Weijun Yu, Ph.D., M.D., M.S.

Academic Editor

PLOS ONE

Additional Editor Comments (optional):

Please address the minor issues raised by Reviewer 2 before finalizing the manuscript for publication.

Reviewers' comments:

Reviewer's Responses to Questions

**Comments to the Author**

1. If the authors have adequately addressed your comments raised in a previous round of review and you feel that this manuscript is now acceptable for publication, you may indicate that here to bypass the “Comments to the Author” section, enter your conflict of interest statement in the “Confidential to Editor” section, and submit your "Accept" recommendation.

Reviewer #1: All comments have been addressed

Reviewer #2: All comments have been addressed

Reviewer #3: All comments have been addressed

2. Is the manuscript technically sound, and do the data support the conclusions?

Reviewer #1: Yes

Reviewer #2: Yes

Reviewer #3: Yes

3. Has the statistical analysis been performed appropriately and rigorously? 

Reviewer #1: Yes

Reviewer #2: Yes

Reviewer #3: Yes

4. Have the authors made all data underlying the findings in their manuscript fully available?

Reviewer #1: Yes

Reviewer #2: Yes

Reviewer #3: Yes

5. Is the manuscript presented in an intelligible fashion and written in standard English?

Reviewer #1: Yes

Reviewer #2: Yes

Reviewer #3: Yes

6. Review Comments to the Author

Reviewer #1: I am happy with the changes made in response to my feedback.

Reviewer #2: The authors have answered my questions and have also produced a work that is clearer and more complete than the previous one.

In particular, they focused solely on linear regression models, which allowed for greater clarity in the exposition. They also conducted various diagnostic analyses, which they rigorously reported in the appendix.

I do not find any reference to the code used for the analyses presented in the paper, but the authors state that it will be made available, presumably through a link in the paper.

Please consider the following issues before publication.

Check formula (1): The range of the indexes is not specified. For example, is it j = 1, 2, 3? Also, σₖ is not defined in the formula, whereas σₗ is present. The subscript s is unclear because the words inside the definition are not properly spaced. However, if the notation is mathematical, it must be preserved and remain consistent, avoiding the insertion of text within formulas. Additionally, there should be a subscript for both the response variable and the model error, as both must refer to the statistical unit if matrix notation is to be avoided. Note that also the model assumptions should also be mentioned.

Formula (2): Same considerations as in the previous point.

Formula (3): Same as above. Moreover, it is unclear why two words are in bold, and ijk is not in italics. The same comment applies to formula (4).

Page 11, line 400: Add "estimated" before "coefficients".

Reviewer #3: (No Response)

7. PLOS authors have the option to publish the peer review history of their article (what does this mean?). If published, this will include your full peer review and any attached files.

Reviewer #1: No

Reviewer #2: No

Reviewer #3: **Yes: **Rodrigo Zepeda Tello

---

## [Editor Report · Acceptance letter]

PONE-D-24-32720R1

PLOS ONE

Dear Dr. Abrar,

I'm pleased to inform you that your manuscript has been deemed suitable for publication in PLOS ONE. Congratulations! Your manuscript is now being handed over to our production team.

Kind regards,

on behalf of

Dr. Weijun Yu

Academic Editor

PLOS ONE